# Learning Boltzmann Generators via Constrained Mass Transport

**Christopher von Klitzing**[1]  **Denis Blessing**[1*]  **Henrik Schopmans**[2*]
**Pascal Friederich**[2]  **Gerhard Neumann**[1]
[1] Autonomous Learning Robots, Karlsruhe Institute of Technology
[2] Artificial Intelligence for Materials Sciences, Karlsruhe Institute of Technology

## Abstract

Efficient sampling from high-dimensional and multimodal unnormalized probability distributions is a central challenge in many areas of science and machine learning. We focus on Boltzmann generators (BGs) that aim to sample the Boltzmann distribution of physical systems, such as molecules, at a given temperature. Classical variational approaches that minimize the reverse Kullback–Leibler divergence are prone to mode collapse, while annealing-based methods, commonly using geometric schedules, can suffer from mass teleportation and rely heavily on schedule tuning. We introduce *Constrained Mass Transport* (CMT), a variational framework that generates intermediate distributions under constraints on both the KL divergence and the entropy decay between successive steps. These constraints enhance distributional overlap, mitigate mass teleportation, and counteract premature convergence. Across standard BG benchmarks and the here introduced *ELIL tetrapeptide*, the largest system studied to date without access to samples from molecular dynamics, CMT consistently surpasses state-of-the-art variational methods, achieving more than 2.5× higher effective sample size while avoiding mode collapse.

## 1 Introduction

We consider the problem of sampling from a target probability measure $p \in \mathcal{P}(\mathbb{R}^d)$ given by $p(x) = \tilde{p}(x)/\mathcal{Z}$ where $\tilde{p} \in C(\mathbb{R}^d, \mathbb{R}_{\geq 0})$ can be evaluated pointwise but the normalization constant $\mathcal{Z} = \int_{\mathbb{R}^d} \tilde{p}(x)\,\mathrm{d}x$ is intractable. Sampling from unnormalized densities arises in many areas, including Bayesian statistics (Gelman et al., 1995), reinforcement learning (Celik et al., 2025), and the natural sciences (Stoltz et al., 2010).

A promising alternative to classical Monte Carlo methods (Hammersley, 2013) is offered by variational approaches (Struwe & Struwe, 2000), which aim to minimize a statistical divergence between a variational probability measure $q \in \mathcal{P}(\mathbb{R}^d)$ and the target $p$, commonly the reverse Kullback–Leibler (KL) divergence

$$q^* = \underset{q \in \mathcal{P}(\mathbb{R}^d)}{\arg\min}\, D_{\mathrm{KL}}(q \,\|\, p), \tag{1}$$

whose unique minimizer is $q^* = p$. A prominent example is the variational learning of molecular Boltzmann generators (BGs) (Noé et al., 2019), for which $\tilde{p}(x) = \exp\left(-E(x)/k_B T\right)$, with $E$ being an energy function, $T$ the temperature, and $k_B$ the Boltzmann constant. BGs enable efficient sampling of thermodynamic ensembles, thereby bypassing costly molecular dynamics (MD) simulations and accelerating the exploration of rare but physically important states. However, learning BGs is challenging as the state space is typically high-dimensional, the target distribution is often highly multimodal, and evaluating $E(x)$ can be very costly, especially when using accurate energies such as those from density-functional theory (Argaman & Makov, 2000).

Furthermore, directly minimizing the reverse KL divergence tends to suffer from mode collapse, ignoring low-probability modes of the target (Blessing et al., 2024; Soletskyi et al., 2025). To address this, a number of recent approaches have proposed to construct a sequence of intermediate

---

*Equal contribution. Correspondence to denis.blessing@kit.edu and henrik.schopmans@kit.edu.

distributions that transport probability mass from a tractable base distribution $q_0$ to the target (Arbel et al., 2021; Matthews et al., 2022; Vargas et al., 2023; Tian et al., 2024; Albergo & Vanden-Eijnden, 2024). This idea, which dates back more than two decades to annealed importance sampling (Neal, 2001), is most often realized through a geometric annealing path, which is defined as a sequence of $(q_i)_{i=1}^I$ which follows $q_i \propto q_0^{1-\beta_i} \tilde{p}^{\beta_i}$, where the corresponding annealing schedule $(\beta_i)_{i=1}^I$ ensures that $q_I = p$. Despite its simplicity, geometric annealing can suffer from mass teleportation, where large portions of the probability mass shift to disjoint regions between successive steps, complicating mass transport (Akhound-Sadegh et al., 2025; Maurais et al., 2025; Chehab & Korba, 2024; Béreux et al., 2024). Moreover, its performance critically depends on the choice of annealing schedule (Syed et al., 2024). By "mass teleportation", we refer to the failure mode of geometric annealing in which probability mass abruptly shifts to regions where the current intermediate has negligible density, blocking effective transport. This differs from the definition of Máté & Fleuret (2023), who focus on preserving relative weights (often termed "mode switching" (Phillips et al., 2024)).

To counteract this, we build on ideas from reinforcement learning (Schulman et al., 2015) and use a trust-region constraint that modifies (1) by bounding the KL divergence between successive distributions, which results in the geometric annealing path with automatic schedule tuning. Adapting these ideas to sampling problems, we further introduce a constraint that explicitly controls the rate at which the entropy of the variational distribution decreases along the transport path. This added degree of freedom enables deviations from the standard geometric annealing path, mitigating issues such as mass teleportation and premature convergence, while fostering greater overlap between consecutive distributions. These contributions culminate in *Constrained Mass Transport* (CMT), a general framework for transporting variational densities along well-defined annealing paths. To highlight its practical utility, we instantiate the framework with normalizing flows and demonstrate that it consistently outperforms state-of-the-art approaches, often by a substantial margin when learning molecular Boltzmann generators directly from energy evaluations, without reliance on additional MD samples.

**Contributions.** Our contributions can be summarized as follows:

- We introduce a general framework for addressing sampling problems through a sequence of constrained variational problems, considering trust-region, entropy, and hybrid constraints.
- We establish a connection between these sequences and annealing paths, which interpolate between a tractable prior and the target distribution.
- We instantiate our framework in practice by employing normalizing flows (Papamakarios et al., 2021) to learn molecular Boltzmann generators (Noé et al., 2019).
- We show that our method, *Constrained Mass Transport* (CMT), consistently surpasses state-of-the-art approaches, often by a significant margin when learning molecular Boltzmann generators solely from energy evaluations, without relying on additional MD samples.
- We introduce a new benchmark, the *ELIL tetrapeptide*, which, to the best of our knowledge, is the largest system studied to date under the setting of learning exclusively from energy evaluations.

## 2 CONSTRAINED MASS TRANSPORT

Here, we denote by $\mathcal{P}(\mathbb{R}^d)$ the space of probability measures on $\mathbb{R}^d$ that are absolutely continuous with respect to Lebesgue measure and admit smooth densities. We approach the sampling problem by dividing (1) into a sequence of constrained optimization problems that result in an annealing path of intermediate densities $(q_i)_{i=0}^I$ that bridge between a tractable prior $q_0$ and the target $p$.

**Trust-region constraint.** Trust regions aim at dividing the problem (1) into subproblems by constraining the updated density to be close to the old density in terms of KL divergence. Formally, this is given by the iterative optimization scheme [1]

$$q_{i+1} = \arg\min_{q \in \mathcal{P}(\mathbb{R}^d)} D_{\mathrm{KL}}(q\|p) \quad \text{s.t.} \quad D_{\mathrm{KL}}(q\|q_i) \leq \varepsilon_{\mathrm{tr}}, \tag{2}$$

---

[1]To ensure that $q \in \mathcal{P}(\mathbb{R}^d)$ we need an additional constraint $\int q(x)\mathrm{d}x = 1$ which we omitted in the main part of the paper for readability.

| Geometric AP linear schedule | Geometric AP via (2) | Tempered AP via (7) | Geometric-tempered AP via (9) |
|---|---|---|---|

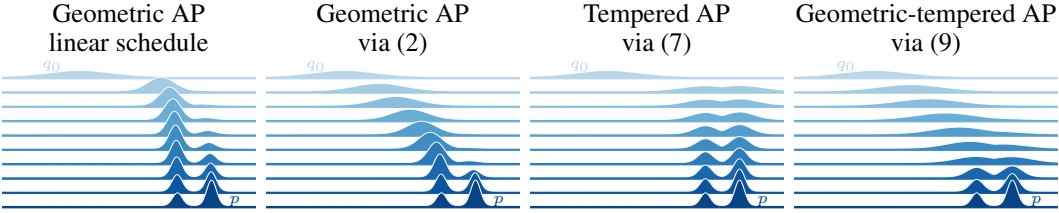

Figure 1: Illustration of the annealing paths (AP) obtained by solving the variational problems (2), (7), or (9). Trust-region–based optimization (2) mitigates the irregularities of naive schedules (e.g., the linear schedule), but the resulting geometric AP suffers from mass teleportation as the right mode of the target distribution $p$ emerges without overlap with earlier intermediate densities. Constraining the entropy decay between successive densities (7) prevents mass teleportation, yet fails to guarantee sufficient overlap between the initial distribution $q_0$ and subsequent intermediate densities. In contrast, combining both constraints (9) yields APs that both maintain overlap between successive densities and avoid mass teleportation.

for $i \in \mathbb{N}$, trust-region bound $\varepsilon_{\text{tr}} > 0$ and some $q_0 \in \mathcal{P}(\mathbb{R}^d)$. Due to the convexity of the KL divergence, we can show that in all but the last step we actually have an equality constraint in (2); see Appendix A. Thus, there exists an $I \in \mathbb{N}$ such that $q_I = q^* (= p)$. Under suitable regularity assumptions, we can approach the above constrained optimization problem using a relaxed Lagrangian formalism, i.e.,

$$\mathcal{L}_{\text{tr}}^{(i+1)}(q, \lambda) = D_{\text{KL}}(q\|p) + \lambda \left(D_{\text{KL}}(q\|q_i) - \varepsilon_{\text{tr}}\right) \tag{3}$$

where $\lambda \geq 0$ is a Lagrangian multiplier, and solve the saddle point problems

$$\max_{\lambda \geq 0} \min_{q \in \mathcal{P}(\mathbb{R}^d)} \mathcal{L}_{\text{tr}}^{(i)}(q, \lambda). \tag{4}$$

We note that $\mathcal{L}_{\text{tr}}^{(i)}$ is convex in $q$ by convexity of the KL divergence and the dual function

$$g_{\text{tr}}^{(i+1)}(\lambda) \coloneqq \min_{q \in \mathcal{P}(\mathbb{R}^d)} \mathcal{L}_{\text{tr}}^{(i)}(q, \lambda)$$

concave in $\lambda$ since it is the pointwise minimum of a family of linear functions of $\lambda$. Thus, (4) has unique optima which we denote by $q_{i+1}$ and $\lambda_i$, respectively. Indeed, (2) admits an analytical solution which is characterized by Proposition 2.1. We refer to Appendix A for a proof and further details on problem (2).

**Proposition 2.1** (Optimal intermediate trust-region densities). *The intermediate optimal densities that solve* (2) *satisfy*

$$q_{i+1}(x, \lambda) = \frac{q_i(x)^{\frac{\lambda}{1+\lambda}} \tilde{p}(x)^{\frac{1}{1+\lambda}}}{\mathcal{Z}_{i+1}(\lambda)}, \quad with \quad \mathcal{Z}_{i+1}(\lambda) = \int q_i(x)^{\frac{\lambda}{1+\lambda}} \tilde{p}(x)^{\frac{1}{1+\lambda}} \, \mathrm{d}x, \tag{5}$$

*where $q_{i+1}$ are the unique optima of the Lagrangian corresponding to* (2).

The optimal multiplier $\lambda_i$ that solves (2) is obtained by plugging $q_{i+1}(\lambda)$ in the Lagrangian (3) to obtain the dual function $g_{\text{tr}} \in C(\mathbb{R}, \mathbb{R})$ given by

$$g_{\text{tr}}^{(i+1)}(\lambda) \coloneqq \mathcal{L}_{\text{tr}}^{(i+1)}(q_{i+1}(\lambda), \lambda) = -(1+\lambda) \log \mathcal{Z}_{i+1}(\lambda) - \lambda \varepsilon_{\text{tr}}. \tag{6}$$

Assuming access to $\mathcal{Z}_{i+1}(\lambda)$ one can solve $\lambda_i = \arg\max_{\lambda \geq 0} g_{\text{tr}}^{(i+1)}(\lambda)$ to obtain the optimal $q \in \mathcal{P}(\mathbb{R}^d)$ that solves (2) as $q_{i+1} \coloneqq q_{i+1}(\lambda_i)$.

**Entropy constraint.** In a similar fashion to (2), we can avoid premature convergence by regulating the entropy decay of the model by constructing a sequence of intermediate densities whose change in entropy is constrained. Formally, we aim to solve the following problem

$$q_{i+1} = \arg\min_{q \in \mathcal{P}(\mathbb{R}^d)} D_{\text{KL}}(q\|p) \quad \text{s.t.} \quad H(q_i) - H(q) \leq \varepsilon_{\text{ent}}, \tag{7}$$

where $H(q) = -\int q(x) \log q(x) \mathrm{d}x$ is the Shannon entropy and $\varepsilon_{\text{ent}} > 0$ the entropy bound. We can again approach (7) using a Lagrangian formalism by introducing a Lagrangian multiplier $\eta \geq 0$. The analytical solution to (7) is characterized by Proposition 2.2 whose proof can be found in Appendix A.

**Proposition 2.2** (Optimal intermediate densities for entropy constraint). *The intermediate optimal densities that solve* (7) *satisfy*

$$q_{i+1}(x, \eta) = \frac{\tilde{p}(x)^{\frac{1}{1+\eta}}}{\mathcal{Z}_{i+1}(\eta)}, \quad with \quad \mathcal{Z}_{i+1}(\eta) = \int \tilde{p}(x)^{\frac{1}{1+\eta}} \, \mathrm{d}x, \tag{8}$$

*where $q_{i+1}$ are the unique optima of the Lagrangian corresponding to* (7).

Despite the potential of (7) for counteracting premature convergence, we identify two challenges depending on the entropy of the initial density $H(q_0)$: First, if $H(q_0) < H(p)$ then the constraint is inactive resulting in $\eta_0 = 0$, reducing (7) to the optimization problem as stated in (1). Second, if $H(q_0) \gg H(p)$ then the KL divergence between $q_0$ and $q_1 \propto p^{1/1+\eta_0}$ can be arbitrarily large and therefore could cause instabilities due to a lack of overlap between the successive densities. While the former challenge can typically be addressed by initializing $q_0$ with large entropy, the second can be more intricate. In the following, we explain how this challenge can be addressed by combining the trust-region and entropy constraint.

**Combining both constraints.** One can straightforwardly combine the constraints in (2) and (7) into a single iterative optimization scheme defined as

$$q_{i+1} = \underset{q \in \mathcal{P}(\mathbb{R}^d)}{\arg\min} \ D_{\mathrm{KL}}(q\|p) \quad \text{s.t.} \quad \begin{cases} D_{\mathrm{KL}}(q\|q_i) \leq \varepsilon_{\mathrm{tr}}, \\ H(q_i) - H(q) \leq \varepsilon_{\mathrm{ent}}. \end{cases} \tag{9}$$

In analogy to the previous section, we introduce Lagrangian multipliers $\lambda$ and $\eta$ for the trust-region and entropy constraint, respectively. Indeed, one can again obtain an analytical expression for the evolution of the optimal densities, see Proposition 2.3 and Appendix A for a proof.

**Proposition 2.3** (Optimal intermediate densities for entropy and trust-region constraint). *The intermediate optimal densities that solve* (9) *satisfy*

$$q_{i+1}(x, \lambda, \eta) = \frac{q_i(x)^{\frac{\lambda}{1+\lambda+\eta}} \tilde{p}(x)^{\frac{1}{1+\lambda+\eta}}}{\mathcal{Z}_{i+1}(\lambda, \eta)} \quad with \quad \mathcal{Z}_{i+1}(\lambda, \eta) = \int q_i(x)^{\frac{\lambda}{1+\lambda+\eta}} \tilde{p}(x)^{\frac{1}{1+\lambda+\eta}}(x) \mathrm{d}x, \tag{10}$$

*where $q_{i+1}$ are the unique optima of the Lagrangian corresponding to* (9).

Clearly, if $H(q_0) \gg H(p)$, the trust-region constraint ensures that the KL divergence between $q_0$ and $q_1$ is at most $\varepsilon_{\mathrm{tr}}$ and, therefore, for a suitable choice of $\varepsilon_{\mathrm{tr}}$ ensures that two consecutive densities have sufficient overlap. Lastly, the Lagrangian dual function $g_{\mathrm{tr-ent}} \in C(\mathbb{R}^2, \mathbb{R})$ corresponding to (9), that is,

$$g_{\mathrm{tr-ent}}^{(i+1)}(\lambda, \eta) := -(1 + \lambda + \eta) \log \mathcal{Z}_{i+1}(\lambda, \eta) - \lambda \varepsilon_{\mathrm{tr}} - \eta(H(q_i) - \varepsilon_{\mathrm{ent}}), \tag{11}$$

requires solving a two-dimensional convex optimization problem to obtain $\lambda_i, \eta_i$ which can be done efficiently in practice; see Section 3 for additional details.

**Connection to annealing paths.** Iteratively solving (2), (7) or (9) induces an *annealing path*, that is, a sequence of densities $(q_i)_{i\in\mathbb{N}}$ that interpolates between $q_0$ and $p$. We characterize these paths in Theorem 2.4; See Appendix A for a proof.

**Theorem 2.4** (Annealing paths). *Let $p \in \mathcal{P}(\mathbb{R}^d)$ be the target density and $q_0 \in \mathcal{P}(\mathbb{R}^d)$ some initial density. The intermediate optimal densities that solve* (2), (7) *and* (9) *satisfy*

$$q_i \propto q_0^{1-\beta_i} \tilde{p}^{\beta_i}, \quad q_i \propto \tilde{p}^{\alpha_i} \ (i \geq 1), \quad and \quad q_i \propto q_0^{1-\beta_i} (\tilde{p}^{\alpha_i})^{\beta_i}, \tag{12}$$

*respectively, with $\beta$ and $\alpha$ being functions of the corresponding Lagrangian multipliers. Moreover, the sequences $(\alpha_i)_{i\in\mathbb{N}_0}$ and $(\beta_i)_{i\in\mathbb{N}_0}$ take values in $[0, 1]$, satisfy $\alpha_0 = \beta_0 = 0$ and $\alpha_I = \beta_I = 1$ for some $I \in \mathbb{N}_+$ and $(\beta_i)_{i\in\mathbb{N}_0}$ is monotonically increasing.*

In what follows, we refer to the annealing paths in (12) as geometric (G), tempered (T), and geometric-tempered (GT) annealing paths, respectively. We further refer to Figure 1 for an illustration of the impact of the introduced constraints on the annealing paths.

## 3  LEARNING THE INTERMEDIATE DENSITIES

**A general recipe.**  So far, we discussed how one can construct a sequence of intermediate measures $(q_i)_{i \in \mathbb{N}}$ using our constrained mass transport formulation. However, despite having access to the analytical form of $q_i$, it is typically not possible to sample from it directly. As such, we approximate each $q_i$ by a distribution from a tractable class $\mathcal{Q} \subset \mathcal{P}(\mathbb{R}^d)$ that permits efficient sampling and density evaluation. Given an approximation family $\mathcal{Q}$, we select $\hat{q}_i \in \mathcal{Q}$ to approximate $q_i$ by solving

$$\hat{q}_i = \arg\min_{q \in \mathcal{Q}} D(q_i, q), \tag{13}$$

where $D$ is an arbitrary statistical divergence between probability measures. This formulation is general: the choice of $\mathcal{Q}$ and $D$ determines the trade-off between expressivity, computational cost, and statistical properties such as mode coverage or robustness.

**Practical algorithm.**  In this work, we choose $\mathcal{Q}$ to be a normalizing flow family constructed via push-forwards of a simple base measure (Rezende & Mohamed, 2015; Dinh et al., 2016; Durkan et al., 2019; Kingma & Dhariwal, 2018; Gabrié et al., 2022; Kolesnikov et al., 2024; Zhai et al., 2024). Let $q_z \in \mathcal{P}(\mathbb{R}^d)$ be an easy-to-sample base measure (e.g., a standard Gaussian), and let $\mathcal{F}$ be a class of smooth invertible maps $f : \mathbb{R}^d \to \mathbb{R}^d$. We define

$$\mathcal{Q}_{\mathrm{NF}} := \{ f_\# q_z \mid f \in \mathcal{F} \}, \quad \text{with} \quad (f_\# q_z)(z) = q_z\big(f^{-1}(z)\big) \left| \det \frac{\partial f^{-1}(z)}{\partial z} \right|. \tag{14}$$

with push-forward $f_\# q_z$. To fit $\hat{q}_i$ within this family, we take $D$ to be the importance-weighted forward KL divergence

$$\hat{q}_{i+1} = \arg\min_{q \in \mathcal{Q}_{\mathrm{NF}}} D_{\mathrm{KL}}(q_{i+1} \| q) \quad \text{with} \quad D_{\mathrm{KL}}(q_{i+1} \| q) = \mathbb{E}_{x \sim q_i} \left[ \frac{q_{i+1}(x)}{q_i(x)} \log \left( \frac{q_{i+1}(x)}{q(x)} \right) \right] \tag{15}$$

This choice offers several advantages. First, forward KL strongly penalizes underestimating the support of $q_{i+1}$, encouraging mode coverage and reducing the risk of mode collapse. Second, because $q_{i+1}$ is available in closed form from the constrained transport updates (see Proposition 2.1, 2.2 and 2.3), the importance weights $q_{i+1}(x)/q_i(x)$ can be computed solely from $q_i$ and $\tilde{p}$. Third, the importance-weighted formulation allows us to reuse samples drawn from $q_i$, enabling a seamless integration of replay buffers, resulting in increased sample efficiency. Lastly, the trust-region constraint controls the variance of the importance weights, keeping it approximately constant, independent of the problem dimension $d$ (see Appendix C.3), resulting in a highly scalable algorithm. Details regarding this specific form of CMT are provided in Appendix C.2.

**Lagrangian dual optimization.**  Maximizing the concave dual function (11) requires evaluating intermediate normalization constants $\mathcal{Z}_{i+1}$. This can be done efficiently by expressing $\mathcal{Z}_{i+1}$ as an expectation under $q_i$ and using Monte Carlo estimation. For instance, the expression for $\mathcal{Z}_{i+1}$ in (10) can be estimated as

$$\mathcal{Z}_{i+1}(\lambda, \eta) = \mathbb{E}_{x \sim q_i} \left[ \left( \frac{\tilde{p}(x)}{q_i(x)^{1+\eta}} \right)^{\frac{1}{1+\lambda+\eta}} \right] \approx \frac{1}{N} \sum_{x_n \sim q_i} \left( \frac{\tilde{p}(x_i)}{q_i(x_i)^{1+\eta}} \right)^{\frac{1}{1+\lambda+\eta}}. \tag{16}$$

We note that, in contrast to estimating the normalization constant of the target $\mathcal{Z}$, estimation of the normalization constant of the intermediate distributions $\mathcal{Z}_{i+1}$ can be performed with low variance since the trust-region constraint ensures sufficient overlap with the next intermediate distribution. Furthermore, samples $x_n \sim q_i$ and the corresponding evaluations $q_i(x_n)$ and $\tilde{p}(x_n)$ are typically already computed when solving (13), so the additional cost of determining the Lagrangian multipliers is negligible (see Appendix D.4). For example, on alanine dipeptide, it accounts for only about $0.01\%$ of the total training time. Details of the dual optimization procedure are provided in Appendix C.1, including a code example. Lastly, we refer to Algorithm 1 for an algorithmic overview of the trained measure transport method.

## 4  RELATED WORK

**Boltzmann generators.**  Learning molecular Boltzmann generators (Noé et al., 2019) purely from energy evaluations has been explored with both flow-based methods (Stimper et al., 2022b; Midgley

---

**Algorithm 1** Constrained mass transport

---

**Require:** Initial measure $q_0$, target measure $\tilde{p}$, divergence $D$, approximation family $\mathcal{Q}$, buffer size $N$
    **for** $i \leftarrow 0, \ldots, I-1$ **do**
        Draw $N$ samples $x_n \sim q_i$, evaluate $q_i(x_n), \tilde{p}(x_n)$
        Initialize buffer $\mathcal{B}^{(i)} = (x_n, q_i(x_n), \tilde{p}(x_n))_{n=1}^{N}$
        Compute multipliers $\lambda_i, \eta_i = \arg\max_{\lambda, \eta \in \mathbb{R}^+} g_{\text{tr-ent}}^{(i+1)}(\lambda, \eta)$ using $\mathcal{B}^{(i)}$
        Compute $q_{i+1} \approx \hat{q}_{i+1} = \arg\min_{q \in \mathcal{Q}} D(q_{i+1}, q)$ using $\mathcal{B}^{(i)}$
    **return** $\hat{q}_I \approx p$

---

et al., 2022; Schopmans & Friederich, 2025) and diffusion-based methods (Liu et al., 2025; Choi et al., 2025; Kim et al., 2025). While flow-based approaches have demonstrated strong performance, their diffusion-based counterparts remain less competitive on molecular systems, often struggling with mode collapse, even on relatively small systems.

Next to purely energy-based approaches, recent work showed success in leveraging samples from a higher temperature, which are typically easier to obtain, and transferring the distribution to a lower target temperature (Dibak et al., 2022a; Wahl et al., 2025; Schopmans & Friederich, 2025; Rissanen et al., 2025; Akhound-Sadegh et al., 2025).

Alternative methods train on samples from molecular dynamics (Klein et al., 2023; Midgley et al., 2023; Tan et al., 2025a; Peng & Gao, 2025), allowing for amortized sampling due to transferability to unseen systems (Jing et al., 2022; Abdin & Kim, 2023; Klein & Noé, 2024; Jing et al., 2024; Lewis et al., 2025; Tan et al., 2025b).

**Constrained optimization.** Trust-region methods have a long history as robust optimization algorithms that iteratively minimize an objective within an adaptively sized "trust region"; see Conn et al. (2000) for an overview. Beyond classical optimization, these methods have been extended to operate over spaces of probability distributions, with applications in reinforcement learning (Peters et al., 2010; Schulman et al., 2015; 2017; Achiam et al., 2017; Pajarinen et al., 2019; Akrour et al., 2019; Yang et al., 2020; Otto et al., 2021; Xu et al., 2024; Wu et al., 2017; Abdolmaleki et al., 2018b;a; Meng et al., 2021), black-box optimization (Sun et al., 2009; Wierstra et al., 2014; Abdolmaleki et al., 2015), variational inference (Arenz et al., 2020; 2022), and path integral control (Gómez et al., 2014; Thalmeier et al., 2020). The first explicit link between trust-region optimization and geometric annealing paths was established by Blessing et al. (2025) for path space measures in the setting of stochastic optimal control. Entropy constraints, often introduced as entropy regularization, have also been studied in policy optimization and reinforcement learning, either in the form of soft constraints (Ahmed et al., 2019; Mnih et al., 2016; O'Donoghue et al., 2016) or hard constraints (Abdolmaleki et al., 2015; Pajarinen et al., 2019; Akrour et al., 2016; 2018; 2019). However, prior work typically constrains the absolute entropy value, which is problematic for inference tasks, since it requires prior knowledge of the target density's entropy. To the best of our knowledge, such methods have not yet been extended to sampling problems. Furthermore, the connection between entropy-constrained optimization and annealing paths has not previously been established.

**Improved annealing paths.** Research on improving annealing paths (APs) has largely focused on geometric APs in the context of annealed importance sampling (AIS) (Neal, 2001) and their extensions to sequential Monte Carlo (SMC) (Del Moral et al., 2006); see Jasra et al. (2011); Goshtasbpour et al. (2023); Chopin et al. (2023); Syed et al. (2024). Beyond the standard geometric AP, alternative constructions have been proposed, such as the moment-averaging path for exponential family distributions (Grosse et al., 2013) and the arithmetic mean path (Chen et al., 2021). The geometric path itself can be interpreted as a quasi-arithmetic mean (Kolmogorov & Castelnuovo, 1930) under the natural logarithm, which motivated Brekelmans et al. (2020) to propose APs based on the deformed logarithm transformation. A variational characterization of these paths was later analysed by Brekelmans & Nielsen (2024). Related work also explores improved schedules for parallel tempering Surjanovic et al. (2022); Syed et al. (2021) and for the diffusion coefficient in ergodic Ornstein–Uhlenbeck processes used to train denoising diffusion models (Ho et al., 2020;

Song et al., 2020); see, e.g., Nichol & Dhariwal (2021); Williams et al. (2024); Benita et al. (2025); Zhang (2025).

## 5 NUMERICAL EVALUATION

Table 1: Results for all systems of varying dimensionality $d$. Evaluation criteria include the number of target evaluations (TARGET EVALS), the evidence upper bound (EUBO), the reverse effective sample size (ESS) and the average total variation distance to the Ramachandran plots generated from molecular dynamics samples (RAM TV). Details on all metrics can be found in Appendix D.3. Each value is shown as the mean $\pm$ standard error over four independent runs. An exception is TA-BG on the ELIL tetrapeptide, for which only two runs were successful due to numerical instabilities. The best results are highlighted in bold, except for the forward KL and reverse KL. Reverse KL is prone to mode collapse, which makes ESS values not directly comparable, and forward KL is trained from samples rather than from energy.

| SYSTEM | METHOD | TARGET EVALS $\downarrow$ | EUBO $\downarrow$ | ESS [%] $\uparrow$ | RAM TV $\downarrow$ | |
|---|---|---|---|---|---|---|
| ALANINE DIPEPTIDE ($d = 60$) | FORWARD KL | $5 \times 10^9$ | $-174.92 \pm 0.00$ | $(82.14 \pm 0.08)\,\%$ | $(1.09 \pm 0.01) \times 10^{-2}$ | |
| | REVERSE KL | $2.56 \times 10^8$ | $-174.96 \pm 0.00$ | $(94.13 \pm 0.21)\,\%$ | $(1.36 \pm 0.05) \times 10^{-2}$ | |
| | FAB | $2.13 \times 10^8$ | $-174.98 \pm 0.00$ | $(94.80 \pm 0.04)\,\%$ | $(1.03 \pm 0.01) \times 10^{-2}$ | |
| | TA-BG | $1 \times 10^8$ | $-174.99 \pm 0.00$ | $(95.76 \pm 0.13)\,\%$ | $(1.24 \pm 0.07) \times 10^{-2}$ | |
| | CMT (OURS) | $1 \times 10^8$ | $\mathbf{-175.00 \pm 0.00}$ | $\mathbf{(97.69 \pm 0.03)\,\%}$ | $\mathbf{(9.43 \pm 0.08) \times 10^{-3}}$ | |
| ALANINE TETRA-PEPTIDE ($d = 120$) | FORWARD KL | $4.2 \times 10^9$ | $-333.79 \pm 0.00$ | $(45.29 \pm 0.11)\,\%$ | $(1.47 \pm 0.03) \times 10^{-2}$ | |
| | REVERSE KL | $2.56 \times 10^8$ | $-332.96 \pm 0.13$ | $(75.06 \pm 3.50)\,\%$ | $(2.89 \pm 0.02) \times 10^{-2}$ | |
| | FAB | $2.13 \times 10^8$ | $-333.93 \pm 0.00$ | $(63.59 \pm 0.23)\,\%$ | $(3.10 \pm 0.04) \times 10^{-2}$ | |
| | TA-BG | $1 \times 10^8$ | $-333.99 \pm 0.00$ | $(65.81 \pm 0.24)\,\%$ | $(1.53 \pm 0.09) \times 10^{-2}$ | |
| | CMT (OURS) | $1 \times 10^8$ | $\mathbf{-334.00 \pm 0.00}$ | $\mathbf{(68.60 \pm 0.21)\,\%}$ | $\mathbf{(1.43 \pm 0.03) \times 10^{-2}}$ | |
| ALANINE HEXA-PEPTIDE ($d = 180$) | FORWARD KL | $4.2 \times 10^9$ | $-533.16 \pm 0.01$ | $(10.98 \pm 0.11)\,\%$ | $(1.88 \pm 0.01) \times 10^{-2}$ | |
| | REVERSE KL | $2.56 \times 10^8$ | $-529.26 \pm 0.26$ | $(21.83 \pm 1.30)\,\%$ | $(7.73 \pm 0.63) \times 10^{-2}$ | |
| | FAB | $4.2 \times 10^8$ | $-532.98 \pm 0.01$ | $(14.55 \pm 0.05)\,\%$ | $(6.43 \pm 0.03) \times 10^{-2}$ | |
| | TA-BG | $4 \times 10^8$ | $-533.43 \pm 0.00$ | $(18.22 \pm 0.15)\,\%$ | $(2.59 \pm 0.03) \times 10^{-2}$ | |
| | CMT (OURS) | $4 \times 10^8$ | $\mathbf{-533.51 \pm 0.01}$ | $\mathbf{(29.63 \pm 0.08)\,\%}$ | $\mathbf{(2.48 \pm 0.02) \times 10^{-2}}$ | |
| ELIL TETRA-PEPTIDE ($d = 219$) | FORWARD KL | $4.2 \times 10^9$ | $-276.76 \pm 0.00$ | $(5.85 \pm 0.03)\,\%$ | $(1.58 \pm 0.01) \times 10^{-2}$ | |
| | REVERSE KL | $2.56 \times 10^8$ | $-262.34 \pm 3.48$ | $(1.26 \pm 0.53)\,\%$ | $(2.61 \pm 0.27) \times 10^{-1}$ | |
| | FAB | $8.43 \times 10^8$ | $-276.67 \pm 0.01$ | $(7.21 \pm 0.08)\,\%$ | $(7.54 \pm 0.14) \times 10^{-2}$ | |
| | TA-BG | $8 \times 10^8$ | $-277.40 \pm 0.06$ | $(13.75 \pm 1.42)\,\%$ | $\mathbf{(2.54 \pm 0.13) \times 10^{-2}}$ | |
| | CMT (OURS) | $8 \times 10^8$ | $\mathbf{-277.83 \pm 0.00}$ | $\mathbf{(26.06 \pm 0.26)\,\%}$ | $(3.13 \pm 0.03) \times 10^{-2}$ | |

In this section, we compare our approach against state-of-the-art methods on four challenging molecular systems. We provide a brief overview of the experimental setup here, with full details in Appendix D. Additional experimental results are provided in Appendix B, including extended performance metrics, an ablation study on the effect of both constraints, and an analysis of different trust-region bounds across systems of different dimensionality.

### 5.1 EXPERIMENTAL SETUP

**Benchmark problems.** Our evaluation covers a range of molecular systems, beginning with the well-studied alanine dipeptide ($d = 60$) (Dibak et al., 2022b; Stimper et al., 2022b; Midgley et al., 2022; Tan et al., 2025a), and extending to the larger alanine tetrapeptide ($d = 120$) and alanine hexapeptide ($d = 180$), which have only recently been addressed using variational methods (Schopmans & Friederich, 2025). In addition, we introduce a new benchmark, the ELIL tetrapeptide ($d = 219$), which is higher-dimensional and which contains more complex side chain interactions compared to the alanine hexapeptide. To the best of our knowledge, this represents the largest and most complex molecular system investigated using variational approaches to date. A detailed description of all benchmark systems is provided in Appendix D.2, and visualizations of all systems can be found next to Table 1.

While using Lagrangian multipliers as a stopping criterion is possible (as $\lambda = \eta = 0$ implies satisfied constraints), we use a fixed number of annealing steps $\tilde{I}$ to strictly control the computational budget for fair benchmarking (see Algorithm 2).

**Baseline methods.** Our main baselines are Flow Annealed Importance Sampling Bootstrap (FAB) (Midgley et al., 2022) and Temperature-Annealed Boltzmann Generators (TA-BG) (Schopmans & Friederich, 2025), which currently define the state of the art for variational sampling of molecular systems. For reference, we also include reverse and forward KL training; the latter leverages ground truth samples obtained from molecular dynamics (MD) simulations (see Appendix D.2). To ensure a fair comparison, all methods use neural spline flows (Durkan et al., 2019) and identical architectures.

**Performance criteria.** We evaluate methods primarily using three criteria. First, the evidence upper bound (EUBO), computed with ground truth MD samples. Up to an additive constant, the EUBO corresponds to the forward KL divergence and is therefore well suited for detecting mode collapse (Blessing et al., 2024). Second, we consider the effective sample size (ESS), defined as $\mathrm{ESS}(q, p) := \left( \mathbb{E}_{x \sim q} \left[ (p(x)/q(x))^2 \right] \right)^{-1}$. ESS is a common measure of sample quality, but it is known to be less reliable for assessing mode collapse (Blessing et al., 2024).

Finally, we consider Ramachandran plots as a qualitative criterion for assessing mode collapse. These plots visualize the 2D log-density of the joint distribution of a pair of dihedral angles in a peptide's backbone. This is a low-dimensional representation of important molecular configurations, making it possible to assess whether the generated samples capture all relevant modes of the distribution or fail to represent certain regions of the state space. For more details on Ramachandran plots, we refer to Appendix D.3 and Schopmans & Friederich (2025). To assess the quality of the Ramachandran plots, we use the total variation distance (Ram TV) between the model-sampled and ground-truth (MD) Ramachandran histograms, as the TV distance is symmetric and more naturally reflects the bidirectional nature of matching generated and target Boltzmann distributions, thereby also penalizing overestimation of density by the model.

For details on all metrics, we refer to Appendix D.3. Since evaluating the target density of molecular systems is typically expensive, we also report the number of target evaluations required by each method.

## 5.2 RESULTS

**Main results.** The main findings are summarized in Table 1. Across all systems and metrics, our method outperforms the baselines while requiring the same or fewer target evaluations. It produces samples closer to the ground-truth distribution (EUBO), allows more efficient importance sampling (ESS), and provides superior mode coverage and resolution of metastable high-energy regions (RAM TV). While the performance gap between our method and the baselines is less pronounced for smaller systems, it widens substantially for the larger ones. In particular, on alanine hexapeptide and ELIL tetrapeptide, our method attains approximately twice the ESS of competing approaches, while also avoiding mode collapse, as reflected in improved EUBO and Ram TV values. In contrast, the reverse KL objective exhibits significant mode collapse, as evidenced by the widening gap in EUBO and Ram TV relative to the other methods, with the most pronounced discrepancy observed on the largest system, ELIL tetrapeptide.

Taken together, the consistency of these trends across metrics and systems highlights the robustness of our method, particularly in challenging high-dimensional systems.

**Ablation study for constraints.** Additionally, we investigate the effect of different constraint choices on the performance of the alanine hexapeptide system. Specifically, we compare four settings: using both constraints, each constraint individually, and no constraint (which corresponds to importance-weighted forward KL minimization). The results are summarized in Figure 2 and Figure 3. Figure 2a shows that omitting the trust-region constraint causes entropy to decrease rapidly, which leads to mode collapse during training. Moreover, using only the entropy constraint yields unstable training, as evidenced by violations of the prescribed linear entropy decay. In contrast, incorporating a trust-region constraint stabilizes training, as reflected in Figure 2b, where it produces a substantially higher ESS between successive intermediate densities. Figure 3 shows Ramachandran plots of alanine hexapeptide with the constraints selectively enabled or disabled. Visible signs of mode collapse appear in all cases except for the tempered (7) and geometric-tempered (9) variants, with the most accurate Ramachandran plot observed in the latter. Overall, our findings indicate that both constraints are necessary to achieve high ESS values while simultaneously avoiding mode collapse.

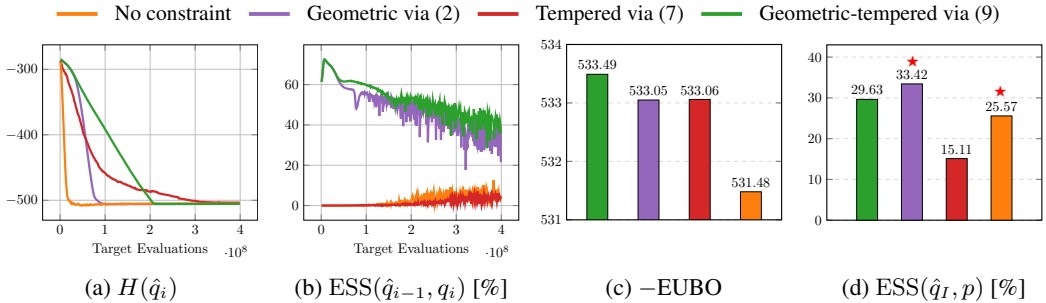

(a) $H(\hat{q}_i)$      (b) $\mathrm{ESS}(\hat{q}_{i-1}, q_i)$ [%]      (c) $-\mathrm{EUBO}$      (d) $\mathrm{ESS}(\hat{q}_I, p)$ [%]

Figure 2: Impact of the trust-region and entropy constraint visualized on alanine hexapeptide. Figure 2a visualizes the model entropy over the course of the training. Analogously, Figure 2b shows the importance-weight effective sample size (ESS) between successive intermediate densities. Figures 2c and 2d depict the final log-likelihood and ESS to the target density, respectively. The variants in Figure 2d marked with "★" exhibit visible mode-collapse on the Ramachandran plots; see Figure 3. The ESS is therefore not directly comparable to the other methods. We denote by $\hat{q}_i$ the variational approximation of the intermediate density $q_i$.

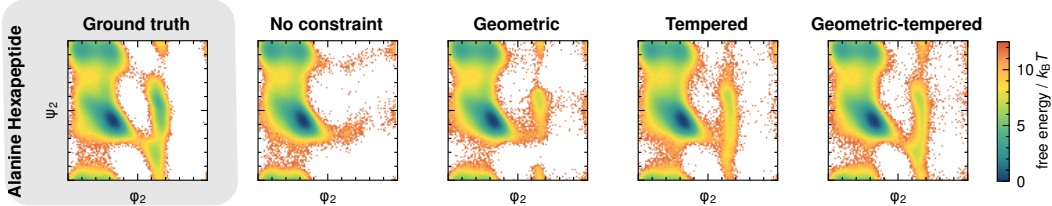

Figure 3: Ramachandran plots for alanine hexapeptide with trust-region and entropy constraints selectively enabled or disabled. Using a single or no constraint leads to mode collapse, whereas combining both constraints avoids it. See Appendix D.3 for details.

## 6 CONCLUSION

We have introduced *Constrained Mass Transport* (CMT), a variational framework for constructing intermediate distributions that transport probability mass from a tractable base measure to a complex, unnormalized target. By enforcing constraints on both the KL divergence and the entropy decay between successive steps, CMT balances exploration and convergence, thereby mitigating mass teleportation, reducing mode collapse, and promoting smooth distributional overlap. Our empirical evaluation across established Boltzmann generator benchmarks and the here proposed *ELIL tetrapeptide*, learned purely from energy evaluations without access to molecular dynamics samples, demonstrates that CMT consistently outperforms existing annealing-based and variational baselines, achieving over $2.5\times$ higher effective sample size while preserving mode diversity.

Promising directions for future work include exploring alternative approximation families $\mathcal{Q}$ and divergences $D$ for learning intermediate densities, which may lead to further performance improvements. Applying our method in Cartesian coordinate representations also presents an interesting avenue, as it facilitates transferability across different molecular systems (Klein & Noé, 2024; Tan et al., 2025b). A key limitation of the current approach is the large number of gradient updates needed to approximate each intermediate target during training. Future work could investigate using more efficient loss functions, such as the log-variance loss (Richter et al., 2020), to reduce computational cost and improve scalability.

## REPRODUCIBILITY STATEMENT

The source code for all experiments is available at `https://github.com/ChristophervonKlitzing/CMT-Molecular`. The ground-truth molecular dynamics data have been made publicly accessible at `https://doi.org/10.5281/zenodo.18822445`.

## ACKNOWLEDGMENTS

The authors acknowledge support by the state of Baden-Württemberg through bwHPC. This work is supported by the Helmholtz Association Initiative and Networking Fund on the HAICORE@KIT partition. D.B. acknowledges support by funding from the pilot program Core Informatics of the Helmholtz Association (HGF). H.S. acknowledges financial support by the German Research Foundation (DFG) through the Research Training Group 2450 "Tailored Scale-Bridging Approaches to Computational Nanoscience". P.F. acknowledges funding from the Klaus Tschira Stiftung gGmbH (SIMPLAIX project) and the pilot program Core-Informatics of the Helmholtz Association (KiKIT project).

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

APPENDIX

# A    PROOFS

*Proof of Propositions 2.1 to 2.3.* We divide the proof into two parts and start with the most general formulation using both constraints, referring to Proposition 2.3. We will then derive the solution with just the trust-region (Proposition 2.1) and just the entropy constraint (Proposition 2.2), as they can be interpreted as special cases of the more general formulation.

**General formulation.**    Consider the constrained optimization problem

$$q_{i+1} = \underset{q \in \mathcal{P}(\mathbb{R}^d)}{\arg\min} D_{\mathrm{KL}}(q\|p) \quad \text{s.t.} \quad D_{\mathrm{KL}}(q\|q_i) \leq \varepsilon_{\mathrm{tr}}, \quad H(q_i) - H(q) \leq \varepsilon_{\mathrm{ent}}, \quad \int \mathrm{d}q = 1 \quad (17)$$

and its corresponding Lagrangian

$$\begin{aligned} \mathcal{L}_{\mathrm{tr}}^{(i+1)}(q, \lambda, \eta, \omega) = D_{\mathrm{KL}}(q\|p) &+ \lambda(D_{\mathrm{KL}}(q\|q_i) - \varepsilon_{\mathrm{tr}}) \\ &+ \eta(H(q_i) - H(q) - \varepsilon_{\mathrm{ent}}) \\ &+ \omega\left(\int \mathrm{d}q - 1\right). \end{aligned} \quad (18)$$

Using the convexity of the KL divergence in its arguments, the convexity of the negative Shannon entropy, and that the integral is a linear functional, the objective from Equation (17) and its Lagrangian, given by Equation (18), are convex in $q$. Using that $\mathcal{P}(\mathbb{R}^d)$ is continuous, there always exists a measure $\tilde{q} \neq q_i$ with $D_{\mathrm{KL}}(\tilde{q}\|q_i) < \varepsilon_{\mathrm{tr}}$ and $H(q_i) - H(\tilde{q}) < \varepsilon_{\mathrm{ent}}$ that satisfies the inequality constraints strictly. Following Boyd & Vandenberghe (2004) (Sec. 5.2.3), Slater's condition holds, implying strong duality. We can therefore instead solve the dual problem.

We start by setting up the Euler-Lagrange equation, given by

$$\frac{\partial}{\partial q}\mathcal{L}_{\mathrm{tr}}^{(i+1)}(q, \lambda, \eta, \omega) = 0,$$

using

$$\begin{aligned} \mathcal{L}_{\mathrm{tr}}^{(i+1)}(q, \lambda, \eta, \omega) = &\int q(x)\big((1 + \lambda + \eta)\log q(x) - \log p(x) - \lambda \log q_i(x) + \omega\big)\mathrm{d}x \\ &- \lambda\varepsilon_{\mathrm{tr}} + \eta(H(q_i) - \varepsilon_{\mathrm{ent}}) - \omega \end{aligned}$$

and solve for $q$. Hence, we get

$$\frac{\partial}{\partial q}\mathcal{L}_{\mathrm{tr}}^{(i+1)}(q, \lambda, \eta, \omega) = (1 + \lambda + \eta)(\log q + 1) - \log p - \lambda \log q_i + \omega = 0$$

$$\Leftrightarrow \log q = \log\left(q_i^{\frac{\lambda}{1+\lambda+\eta}} p^{\frac{1}{1+\lambda+\eta}}\right) - \left(\frac{\omega}{1+\lambda+\eta} + 1\right). \quad (19)$$

Using this, we can further determine $\omega$ using

$$\int \mathrm{d}q = \int q_i(x)^{\frac{\lambda}{1+\lambda+\eta}} p(x)^{\frac{1}{1+\lambda+\eta}}\mathrm{d}x \Big/ \exp\left(\frac{\omega}{1+\lambda+\eta} + 1\right) = 1$$

$$\Leftrightarrow \omega = (1 + \lambda + \eta)(\log \bar{\mathcal{Z}}_{i+1}(\lambda, \eta) - 1), \quad \text{with} \quad \bar{\mathcal{Z}}_{i+1}(\lambda, \eta) = \int q_i(x)^{\frac{\lambda}{1+\lambda+\eta}} p(x)^{\frac{1}{1+\lambda+\eta}}(x)\mathrm{d}x.$$

Substituting $\omega$ back into (19) and simplifying the fraction using $\tilde{p} = \mathcal{Z}p$ yields

$$q_{i+1}(x, \lambda, \eta) = \frac{q_i(x)^{\frac{\lambda}{1+\lambda+\eta}} \tilde{p}(x)^{\frac{1}{1+\lambda+\eta}}}{\mathcal{Z}_{i+1}(\lambda, \eta)} \quad \text{with} \quad \mathcal{Z}_{i+1}(\lambda, \eta) = \int q_i(x)^{\frac{\lambda}{1+\lambda+\eta}} \tilde{p}(x)^{\frac{1}{1+\lambda+\eta}}(x)\mathrm{d}x,$$

which uses the unnormalized target $\tilde{p}$, proving Proposition 2.3.

**Special cases.**    Setting $\varepsilon_{\mathrm{ent}} = \infty$ or $\varepsilon_{\mathrm{tr}} = \infty$ effectively deactivates the respective constraint, yielding the trust-region objective (2) or the entropy objective (7), respectively. This is equivalent to setting the Lagrangian multipliers $\eta = 0$ or $\lambda = 0$, proving Proposition 2.1 and Proposition 2.2, respectively. $\square$

*Proof of Theorem 2.4.* We divide the proof into three parts and start with the most general formulation using both constraints. The first two parts will show form and monotonicity, while part three will derive the special cases with just the trust-region and just the entropy constraint.

**General form (part 1).** Given are the sequences of Lagrangian multipliers $(\lambda_i)_{i\in\mathbb{N}_0} \geq 0$ and $(\eta_i)_{i\in\mathbb{N}_0} \geq 0$. We now aim to proof that the sequence $(q_i)_{i\in\mathbb{N}_0}$, given by

$$\tilde{q}_i = \begin{cases} q_0 & , \quad i = 0 \\ \tilde{q}_{i-1}^{\frac{\lambda_{i-1}}{1+\lambda_{i-1}+\eta_{i-1}}} \tilde{p}^{\frac{1}{1+\lambda_{i-1}+\eta_{i-1}}} & , \quad i \geq 1 \end{cases}, \tag{20}$$

takes the form

$$\tilde{q}_i = q_0^{1-\beta_i}(\tilde{p}^{\alpha_i})^{\beta_i} \quad \text{with} \quad \beta_i = 1 - \prod_{j=0}^{i-1} \frac{\lambda_j}{1+\lambda_j+\eta_j}$$

$$\text{and} \quad \alpha_i = \begin{cases} 0 & , \quad i = 0 \\ 1 - \frac{1}{\beta_i}\sum_{k=0}^{i-1}\frac{\eta_k}{1+\lambda_k+\eta_k}\prod_{j=k+1}^{i-1}\frac{\lambda_j}{1+\lambda_j+\eta_j} & , \quad i \geq 1. \end{cases}$$

We use the common convention that empty products evaluate to one.

**Base case ($i = 0$):** The simplest case

$$\tilde{q}_0 = q_0^{1-\beta_0}(\tilde{p}^{\alpha_0})^{\beta_0}$$

holds due to $\beta_0 = 0$ (using empty product convention).

**Inductive step:** We start from Equation (20) and transform it using the assumption that $\tilde{q}_i = q_0^{1-\beta_i}(\tilde{p}^{\alpha_i})^{\beta_i}$ holds for some arbitrary but fixed $i \in \mathbb{N}_0$, yielding

$$\tilde{q}_{i+1} = \tilde{q}_i^{\frac{\lambda_i}{1+\lambda_i+\eta_i}} \tilde{p}^{\frac{1}{1+\lambda_i+\eta_i}}$$

$$= \left(q_0^{1-\beta_i}(\tilde{p}^{\alpha_i})^{\beta_i}\right)^{\frac{\lambda_i}{1+\lambda_i+\eta_i}} \tilde{p}^{\frac{1}{1+\lambda_i+\eta_i}}$$

$$= q_0^{1-\beta_{i+1}}\tilde{p}^{\alpha_i\beta_i\frac{\lambda_i}{1+\lambda_i+\eta_i}+\frac{1}{1+\lambda_i+\eta_i}}.$$

Using

$$\beta_i\frac{\lambda_i}{1+\lambda_i+\eta_i} = \frac{\lambda_i}{1+\lambda_i+\eta_i} - \prod_{j=0}^{i}\frac{\lambda_j}{1+\lambda_j+\eta_j}$$

$$= 1 - \frac{1+\eta_i}{1+\lambda_i+\eta_i} - \prod_{j=0}^{i}\frac{\lambda_j}{1+\lambda_j+\eta_j}$$

$$= \beta_{i+1} - \frac{1+\eta_i}{1+\lambda_i+\eta_i},$$

we now can rewrite the exponent of $p$ yielding

$$\alpha_i\beta_i\frac{\lambda_i}{1+\lambda_i+\eta_i} + \frac{1}{1+\lambda_i+\eta_i}$$

$$= \left(\beta_i - \sum_{k=0}^{i-1}\frac{\eta_k}{1+\lambda_k+\eta_k}\prod_{j=k+1}^{i-1}\frac{\lambda_j}{1+\lambda_j+\eta_j}\right)\frac{\lambda_i}{1+\lambda_i+\eta_i} + \frac{1}{1+\lambda_i+\eta_i}$$

$$= \beta_{i+1} - \frac{1+\eta_i}{1+\lambda_i+\eta_i} - \sum_{k=0}^{i-1}\frac{\eta_k}{1+\lambda_k+\eta_k}\prod_{j=k+1}^{i}\frac{\lambda_j}{1+\lambda_j+\eta_j} + \frac{1}{1+\lambda_i+\eta_i}$$

$$= \beta_{i+1} - \sum_{k=0}^{i}\frac{\eta_k}{1+\lambda_k+\eta_k}\prod_{j=k+1}^{i}\frac{\lambda_j}{1+\lambda_j+\eta_j},$$

$$= \alpha_{i+1}\beta_{i+1},$$

again using the convention that an empty product evaluates to one. It directly follows

$$q_{i+1} \propto \tilde{q}_{i+1} = q_0^{1-\beta_{i+1}}(\tilde{p}^{\alpha_{i+1}})^{\beta_{i+1}},$$

which completes the induction.

**Monotonicity (part 2).** It remains to show that $(\alpha_i)_{i\in\mathbb{N}_0}$ and $(\beta_i)_{i\in\mathbb{N}_0}$ take values in $[0,1]$ with $\alpha_0 = \beta_0 = 0$ and $\alpha_I = \beta_I = 1$ for some $I \in \mathbb{N}_+$ and $(\beta_i)_{i\in\mathbb{N}_0}$ is monotonically increasing.

The first case ($\alpha_0 = \beta_0 = 0$) holds by definition. Assuming that there exists an $I \in \mathbb{N}_+$, such that $\lambda_{I-1} = \eta_{I-1} = 0$,

$$\beta_i = 1 - \prod_{j=0}^{i-1} \frac{\lambda_j}{1+\lambda_j+\eta_j} \overset{i\geq I}{=} 1$$

and

$$\alpha_i = 1 - \frac{1}{\beta_i} \sum_{k=0}^{i-1} \frac{\eta_k}{1+\lambda_k+\eta_k} \prod_{j=k+1}^{i-1} \frac{\lambda_j}{1+\lambda_j+\eta_j} \overset{i\geq I}{=} 1$$

follow directly for all $i \geq I$. Both the trust-region and entropy constraints become inactive at the optimum and do not prevent $(q_i)_{i\in\mathbb{N}0}$ from reaching the unique optimum $p$ ($\varepsilon_{\mathrm{tr}}, \varepsilon_{\mathrm{ent}} > 0$). Consequently, both Lagrangian multipliers will eventually vanish, motivating the existence of some $I \in \mathbb{N}+$, such that $\lambda_{I-1} = \eta_{I-1} = 0$.

We now show monotonicity of $(\beta_i)_{i\in\mathbb{N}_0}$ using $(\lambda_i)_{i\in\mathbb{N}_0} \geq 0$ and $(\eta_i)_{i\in\mathbb{N}_0} \geq 0$. Let $i \in \mathbb{N}_0$ be arbitrary. As a direct consequence of

$$\beta_{i+1} - \beta_i = \prod_{j=0}^{i-1} \frac{\lambda_j}{1+\lambda_j+\eta_j} - \prod_{j=0}^{i} \frac{\lambda_j}{1+\lambda_j+\eta_j}$$

$$= \left(\prod_{j=0}^{i-1} \frac{\lambda_j}{1+\lambda_j+\eta_j}\right)\left(1 - \frac{\lambda_i}{1+\lambda_i+\eta_i}\right)$$

$$= \left(\prod_{j=0}^{i-1} \frac{\lambda_j}{1+\lambda_j+\eta_j}\right)\left(\frac{1+\eta_i}{1+\lambda_i+\eta_i}\right) \overset{\substack{\lambda_j,\eta_j\geq 0\\ \forall j\in\mathbb{N}_0}}{\geq} 0,$$

the sequence $(\beta_i)_{i\in\mathbb{N}_0}$ must be monotonically increasing.

**Special cases (part 3).** We now consider the special cases in which only the trust-region constraint or the entropy constraint is active by setting the Lagrangian multiplier sequence of the other constraint to zero.

We first consider only the trust-region constraint (2), which corresponds to setting the Lagrangian multiplier of the entropy constraint to zero, i.e., $\eta_i = 0$ for all $i \in \mathbb{N}_0$. In this scenario, $\alpha_i$ simplifies to $\alpha_0 = 0$ and $\alpha_i = 1$ for all $i \geq 1$. Consequently, and using $\beta_0 = 0$, the iterates take the form

$$q_i \propto \tilde{q}_i = q_0^{1-\beta_i}\tilde{p}^{\beta_i}, \quad i \in \mathbb{N}_0,$$

as claimed.

Analogously, the trust-region constraint can be rendered inactive by setting $\lambda_i = 0$ for all $i \in \mathbb{N}_0$, leaving only the entropy constraint active, corresponding to Equation (7). In this case, $\beta_0 = 0$ and $\beta_i = 1$ for all $i \geq 1$, yielding

$$q_i \propto \tilde{q}_i = \begin{cases} q_0 & , \quad i = 0, \\ \tilde{p}^{\alpha_i} & , \quad i \geq 1, \end{cases}$$

which concludes the proof. $\qquad\square$

*Proof of uniqueness and tightness of the trust-region solution.* Closely following Blessing et al. (2025), we now establish the uniqueness of the trust-region solution and show that the trust-region constraint is tight for all but the final step. Specifically, we show

$$D_{\mathrm{KL}}(q_i\|p) < \varepsilon_{\mathrm{tr}} \implies q_i = p$$

$$q_i = \arg\min D_{\mathrm{KL}}(q\|p) \quad \text{s.t.} \quad D_{\mathrm{KL}}(q\|q_{i-1}) \leq \varepsilon_{\mathrm{tr}} \quad \text{is unique}$$

If $D_{\mathrm{KL}}(q_i\|p) < \varepsilon_{\mathrm{tr}}$, the KKT conditions imply that the Lagrangian multiplier satisfies $\lambda_i = 0$, so the constraint is inactive. Consequently, $q_i$ must solve the strictly convex unconstrained problem

$$\min_{q \in \mathcal{P}(\mathbb{R}^d)} D_{\mathrm{KL}}(q\|p),$$

which has the unique minimizer $p$. Since $p$ is feasible ($D_{\mathrm{KL}}(p\|p) = 0 \leq \varepsilon_{\mathrm{tr}}$), it follows that $q_i = p$.

Uniqueness of $q_i$ further follows from the convexity of the feasible set $\{q \in \mathcal{P} \mid D_{\mathrm{KL}}(q\|q_i) \leq \varepsilon_{\mathrm{tr}}\}$ together with the strict convexity of the objective in $q$ when $p$ is fixed. $\qquad\square$

# B  EXTENDED NUMERICAL EVALUATION

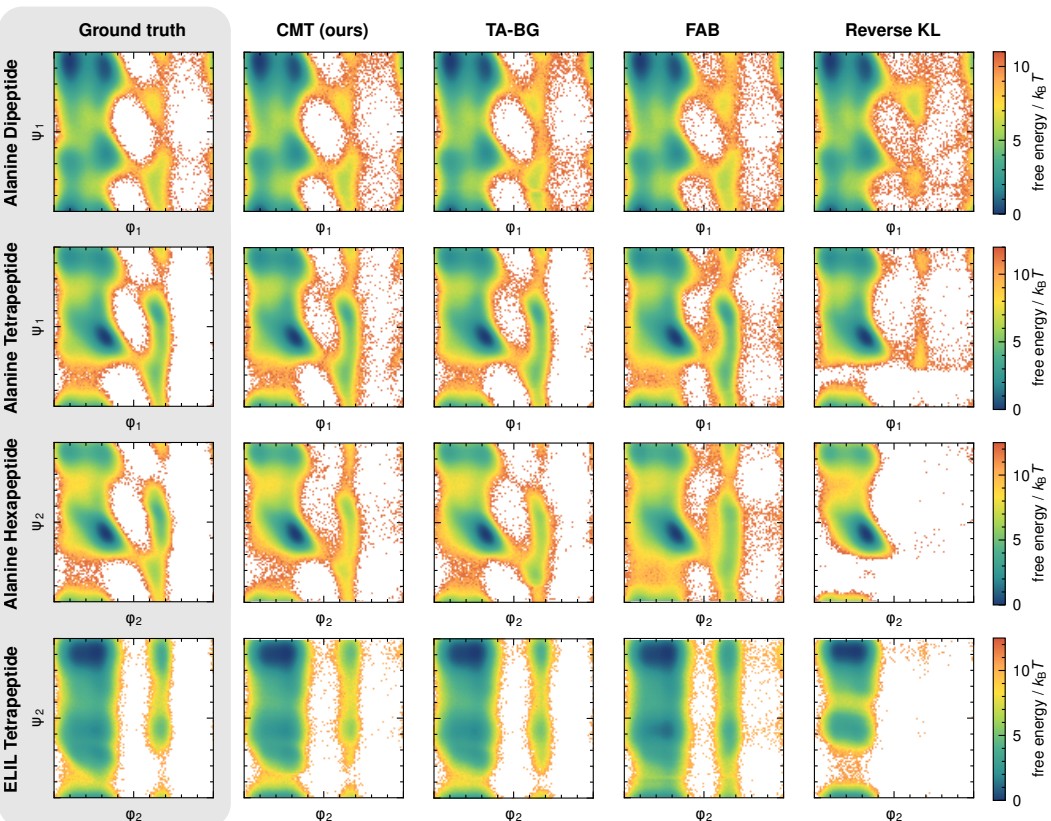

Figure 4: Comparison of Ramachandran plots of backbone dihedral angle pairs obtained with different methods. See Appendix D.3 for details on Ramachandran plots.

Complementing the results of Section 5, this section reports additional metrics for the main method comparison in Table 2 (see also the corresponding Ramachandran plots in Figure 4), an ablation study on the effect of both constraints (see Table 3 and Figure 5), and an ablation study on the trust-region constraint and its effect on bounding importance-weight variance across different system sizes and trust-region bounds $\varepsilon_{\mathrm{tr}}$ (see Figure 6).

**Main results.**  We begin with Table 2, which introduces several additional metrics, such as Ram KL and Ram KL w. RW, to quantify the discrepancy between the ground truth and method-generated Ramachandran plots (see Appendix D.3 for details on all metrics). Substantially elevated Ram KL values serve as robust indicators of mode collapse, as exemplified by the results for reverse KL training, where the Ram KL values are consistently at least an order of magnitude higher than those observed for other methods. Corresponding Ramachandran plots for the different methods are shown in Figure 4. We include Ram KL in our evaluation, despite its asymmetry, to ensure comparability with previous work (Midgley et al., 2022; Schopmans & Friederich, 2025). For the same reason, we

Table 2: Comparison of metrics obtained for all four peptide systems. To remain comparable to FAB (Midgley et al., 2022) and TA-BG (Schopmans & Friederich, 2025), we first report the number of target evaluations (TARGET EVALS), the negative log-likelihood (NLL), the reverse effective sample size (ESS), the average forward KL divergence to the Ramachandran plots (RAM KL) generated from molecular dynamics (MD) samples, and its importance-weighted version (RAM KL W. RW). We then report the evidence lower bound (ELBO), the estimated target log normalization constant ($\log \mathcal{Z}$), the evidence upper bound (EUBO), the average total variation distance to the Ramachandran plots from MD samples and its importance-weighted version, and the torus Wasserstein-2 distance (RAM $\mathbb{T}$-$\mathcal{W}_2$). The torus Wasserstein-2 distance was computed using only $10^4$ samples due to its high computational cost, making it less reliable. All values are presented as the mean $\pm$ standard error across four independent experiments. An exception is TA-BG on the ELIL tetrapeptide, for which only two runs were successful due to numerical instabilities. The best-performing variational method for each metric is highlighted in bold. Reverse KL was excluded, as it tends to suffer from mode collapse, making the corresponding metrics incomparable. The ELBO is marked with "★" because rare extreme log-importance weights can dominate its mean, making it unreliable for comparing Boltzmann generators (see Appendix D.3 for details).

| SYSTEM | METHOD | TARGET EVALS ↓ | NLL ↓ | ESS [%] ↑ | RAM KL ↓ | RAM KL W. RW ↓ |
|---|---|---|---|---|---|---|
| ALANINE DIPEPTIDE ($d=60$) | FORWARD KL | $5 \times 10^9$ | $-214.358 \pm 0.000$ | $(82.14 \pm 0.08)\,\%$ | $(2.17 \pm 0.05) \times 10^{-3}$ | $(1.89 \pm 0.06) \times 10^{-3}$ |
| | REVERSE KL | $2.56 \times 10^8$ | $-214.398 \pm 0.001$ | $(94.13 \pm 0.21)\,\%$ | $(1.66 \pm 0.25) \times 10^{-2}$ | $(1.57 \pm 0.26) \times 10^{-2}$ |
| | FAB | $2.13 \times 10^8$ | $-214.410 \pm 0.000$ | $(94.80 \pm 0.04)\,\%$ | $(1.52 \pm 0.04) \times 10^{-3}$ | $\mathbf{(1.23 \pm 0.01) \times 10^{-3}}$ |
| | TA-BG | $1 \times 10^8$ | $-214.419 \pm 0.001$ | $(95.76 \pm 0.13)\,\%$ | $(1.92 \pm 0.07) \times 10^{-3}$ | $(1.37 \pm 0.02) \times 10^{-3}$ |
| | CMT (OURS) | $1 \times 10^8$ | $\mathbf{-214.429 \pm 0.000}$ | $\mathbf{(97.69 \pm 0.03)\,\%}$ | $\mathbf{(1.48 \pm 0.03) \times 10^{-3}}$ | $(1.35 \pm 0.03) \times 10^{-3}$ |
| ALANINE TETRA-PEPTIDE ($d=120$) | FORWARD KL | $4.2 \times 10^9$ | $-332.020 \pm 0.001$ | $(45.29 \pm 0.11)\,\%$ | $(2.33 \pm 0.06) \times 10^{-3}$ | $(2.33 \pm 0.04) \times 10^{-3}$ |
| | REVERSE KL | $2.56 \times 10^8$ | $-331.189 \pm 0.126$ | $(75.06 \pm 3.50)\,\%$ | $(3.05 \pm 0.35) \times 10^{-1}$ | $(2.93 \pm 0.40) \times 10^{-1}$ |
| | FAB | $2.13 \times 10^8$ | $-332.158 \pm 0.002$ | $(63.59 \pm 0.23)\,\%$ | $(7.00 \pm 0.25) \times 10^{-3}$ | $\mathbf{(1.13 \pm 0.01) \times 10^{-3}}$ |
| | TA-BG | $1 \times 10^8$ | $-332.219 \pm 0.003$ | $(65.81 \pm 0.24)\,\%$ | $\mathbf{(1.92 \pm 0.08) \times 10^{-3}}$ | $(1.56 \pm 0.03) \times 10^{-3}$ |
| | CMT (OURS) | $1 \times 10^8$ | $\mathbf{-332.231 \pm 0.002}$ | $\mathbf{(68.60 \pm 0.21)\,\%}$ | $(2.17 \pm 0.09) \times 10^{-3}$ | $(1.57 \pm 0.09) \times 10^{-3}$ |
| ALANINE HEXA-PEPTIDE ($d=180$) | FORWARD KL | $4.2 \times 10^9$ | $-504.528 \pm 0.005$ | $(10.98 \pm 0.11)\,\%$ | $(4.09 \pm 0.23) \times 10^{-3}$ | $(7.49 \pm 0.06) \times 10^{-3}$ |
| | REVERSE KL | $2.56 \times 10^8$ | $-500.625 \pm 0.265$ | $(21.83 \pm 1.30)\,\%$ | $(5.14 \pm 0.36) \times 10^{-1}$ | $(5.07 \pm 0.35) \times 10^{-1}$ |
| | FAB | $4.2 \times 10^8$ | $-504.346 \pm 0.009$ | $(14.55 \pm 0.05)\,\%$ | $(2.16 \pm 0.02) \times 10^{-2}$ | $(1.06 \pm 0.02) \times 10^{-2}$ |
| | TA-BG | $4 \times 10^8$ | $-504.792 \pm 0.005$ | $(18.22 \pm 0.15)\,\%$ | $\mathbf{(6.33 \pm 0.12) \times 10^{-3}}$ | $\mathbf{(6.27 \pm 0.10) \times 10^{-3}}$ |
| | CMT (OURS) | $4 \times 10^8$ | $\mathbf{-504.875 \pm 0.008}$ | $\mathbf{(29.63 \pm 0.08)\,\%}$ | $(1.18 \pm 0.05) \times 10^{-2}$ | $(1.15 \pm 0.01) \times 10^{-2}$ |
| ELIL TETRA-PEPTIDE ($d=219$) | FORWARD KL | $4.2 \times 10^9$ | $-601.197 \pm 0.005$ | $(5.85 \pm 0.03)\,\%$ | $(2.68 \pm 0.04) \times 10^{-3}$ | $(8.98 \pm 0.07) \times 10^{-3}$ |
| | REVERSE KL | $2.56 \times 10^8$ | $-586.781 \pm 3.485$ | $(1.26 \pm 0.53)\,\%$ | $(1.32 \pm 0.34) \times 10^{0}$ | $(1.27 \pm 0.38) \times 10^{0}$ |
| | FAB | $8.43 \times 10^8$ | $-601.105 \pm 0.009$ | $(7.21 \pm 0.08)\,\%$ | $(2.49 \pm 0.10) \times 10^{-2}$ | $(1.25 \pm 0.10) \times 10^{-2}$ |
| | TA-BG | $8 \times 10^8$ | $-601.840 \pm 0.064$ | $(13.75 \pm 1.42)\,\%$ | $\mathbf{(6.24 \pm 0.84) \times 10^{-3}}$ | $(6.22 \pm 0.51) \times 10^{-3}$ |
| | CMT (OURS) | $8 \times 10^8$ | $\mathbf{-602.272 \pm 0.004}$ | $\mathbf{(26.06 \pm 0.26)\,\%}$ | $(7.95 \pm 0.11) \times 10^{-3}$ | $\mathbf{(4.79 \pm 0.06) \times 10^{-3}}$ |

| SYSTEM | METHOD | ELBO ↑ ★ | $\leq$ log $\mathcal{Z}$ $\leq$ | EUBO ↓ | RAM TV ↓ | RAM TV W. RW ↓ |
|---|---|---|---|---|---|---|
| ALANINE DIPEPTIDE ($d=60$) | FORWARD KL | $-175.91 \pm 0.37$ | $-175.010 \pm 0.000$ | $-174.92 \pm 0.00$ | $(1.09 \pm 0.01) \times 10^{-2}$ | $(9.13 \pm 0.05) \times 10^{-3}$ |
| | REVERSE KL | $-175.34 \pm 0.10$ | $-175.010 \pm 0.000$ | $-174.96 \pm 0.00$ | $(1.36 \pm 0.05) \times 10^{-2}$ | $(9.36 \pm 0.05) \times 10^{-3}$ |
| | FAB | $-335.75 \pm 21.27$ | $-175.010 \pm 0.000$ | $-174.98 \pm 0.00$ | $(1.03 \pm 0.01) \times 10^{-2}$ | $(8.64 \pm 0.08) \times 10^{-3}$ |
| | TA-BG | $-175.04 \pm 0.00$ | $-175.009 \pm 0.000$ | $-174.99 \pm 0.00$ | $(1.24 \pm 0.07) \times 10^{-2}$ | $(8.67 \pm 0.10) \times 10^{-3}$ |
| | CMT (OURS) | $-175.05 \pm 0.01$ | $-175.009 \pm 0.000$ | $\mathbf{-175.00 \pm 0.00}$ | $\mathbf{(9.43 \pm 0.08) \times 10^{-3}}$ | $\mathbf{(8.51 \pm 0.06) \times 10^{-3}}$ |
| ALANINE TETRA-PEPTIDE ($d=120$) | FORWARD KL | $-525.18 \pm 44.70$ | $-334.193 \pm 0.000$ | $-333.79 \pm 0.00$ | $(1.47 \pm 0.03) \times 10^{-2}$ | $(1.04 \pm 0.01) \times 10^{-2}$ |
| | REVERSE KL | $-334.40 \pm 0.01$ | $-334.233 \pm 0.007$ | $-332.96 \pm 0.13$ | $(2.89 \pm 0.02) \times 10^{-2}$ | $(2.75 \pm 0.04) \times 10^{-2}$ |
| | FAB | $-17658.25 \pm 3804.78$ | $-334.193 \pm 0.000$ | $-333.93 \pm 0.00$ | $(3.10 \pm 0.04) \times 10^{-2}$ | $\mathbf{(8.85 \pm 0.04) \times 10^{-3}}$ |
| | TA-BG | $-334.74 \pm 0.07$ | $-334.193 \pm 0.001$ | $-333.99 \pm 0.00$ | $(1.53 \pm 0.09) \times 10^{-2}$ | $(9.29 \pm 0.14) \times 10^{-3}$ |
| | CMT (OURS) | $-399.34 \pm 15.76$ | $-334.193 \pm 0.000$ | $\mathbf{-334.00 \pm 0.00}$ | $\mathbf{(1.43 \pm 0.03) \times 10^{-2}}$ | $(9.65 \pm 0.27) \times 10^{-3}$ |
| ALANINE HEXA-PEPTIDE ($d=180$) | FORWARD KL | $-86802.43 \pm 1275.91$ | $-534.412 \pm 0.002$ | $-533.16 \pm 0.01$ | $(1.88 \pm 0.01) \times 10^{-2}$ | $(1.93 \pm 0.01) \times 10^{-2}$ |
| | REVERSE KL | $-535.46 \pm 0.20$ | $-534.547 \pm 0.009$ | $-529.26 \pm 0.26$ | $(7.73 \pm 0.63) \times 10^{-2}$ | $(5.41 \pm 0.41) \times 10^{-2}$ |
| | FAB | $-417884.37 \pm 11132.83$ | $-534.420 \pm 0.003$ | $-532.98 \pm 0.01$ | $(6.43 \pm 0.03) \times 10^{-2}$ | $(2.14 \pm 0.01) \times 10^{-2}$ |
| | TA-BG | $-51042.60 \pm 4923.09$ | $-534.416 \pm 0.000$ | $-533.43 \pm 0.00$ | $(2.59 \pm 0.03) \times 10^{-2}$ | $\mathbf{(1.85 \pm 0.01) \times 10^{-2}}$ |
| | CMT (OURS) | $-56239.16 \pm 5690.68$ | $-534.428 \pm 0.001$ | $\mathbf{-533.51 \pm 0.01}$ | $\mathbf{(2.48 \pm 0.02) \times 10^{-2}}$ | $(1.97 \pm 0.01) \times 10^{-2}$ |
| ELIL TETRA-PEPTIDE ($d=219$) | FORWARD KL | $-75532.86 \pm 9545.42$ | $-278.483 \pm 0.000$ | $-276.76 \pm 0.00$ | $(1.58 \pm 0.01) \times 10^{-2}$ | $(2.72 \pm 0.03) \times 10^{-2}$ |
| | REVERSE KL | $-282.53 \pm 0.59$ | $-279.079 \pm 0.156$ | $-262.34 \pm 3.48$ | $(2.61 \pm 0.27) \times 10^{-1}$ | $(1.67 \pm 0.27) \times 10^{-1}$ |
| | FAB | $-544538.24 \pm 16286.19$ | $-278.479 \pm 0.002$ | $-276.67 \pm 0.01$ | $(7.54 \pm 0.14) \times 10^{-2}$ | $(3.38 \pm 0.10) \times 10^{-2}$ |
| | TA-BG | $-23950.58 \pm 4259.44$ | $-278.480 \pm 0.002$ | $-277.40 \pm 0.06$ | $\mathbf{(2.54 \pm 0.13) \times 10^{-2}}$ | $(2.18 \pm 0.11) \times 10^{-2}$ |
| | CMT (OURS) | $-990.75 \pm 164.10$ | $-278.481 \pm 0.001$ | $\mathbf{-277.83 \pm 0.00}$ | $(3.13 \pm 0.03) \times 10^{-2}$ | $\mathbf{(1.68 \pm 0.01) \times 10^{-2}}$ |

also report the negative log-likelihood (NLL), which is equivalent to the EUBO and forward KL up to an additive constant. Further details on all metrics are provided in Appendix D.3.

Table 3: Performance of CMT with the trust-region and entropy constraints selectively enabled or disabled.

| System | Constraint | | Target Evals ↓ | EUBO ↓ | Ram KL ↓ | Ram KL w. RW ↓ | Ram TV ↓ | Ram TV w. RW ↓ |
|---|---|---|---|---|---|---|---|---|
| | Trust-Region | Entropy | | | | | | |
| Alanine Dipeptide ($d = 60$) | ✗ | ✗ | $1 \times 10^8$ | $-175.00 \pm 0.00$ | $(1.47 \pm 0.02) \times 10^{-3}$ | $(1.46 \pm 0.01) \times 10^{-3}$ | $(9.13 \pm 0.05) \times 10^{-3}$ | $(8.65 \pm 0.05) \times 10^{-3}$ |
| | ✓ | ✗ | $1 \times 10^8$ | $-175.00 \pm 0.00$ | $(1.45 \pm 0.03) \times 10^{-3}$ | $(1.38 \pm 0.02) \times 10^{-3}$ | $(9.16 \pm 0.06) \times 10^{-3}$ | $(8.59 \pm 0.02) \times 10^{-3}$ |
| | ✗ | ✓ | $1 \times 10^8$ | $-175.00 \pm 0.00$ | $(1.48 \pm 0.01) \times 10^{-3}$ | $(1.29 \pm 0.02) \times 10^{-3}$ | $(9.58 \pm 0.04) \times 10^{-3}$ | $(8.54 \pm 0.04) \times 10^{-3}$ |
| | ✓ | ✓ | $1 \times 10^8$ | $-175.00 \pm 0.00$ | $(1.48 \pm 0.03) \times 10^{-3}$ | $(1.35 \pm 0.03) \times 10^{-3}$ | $(9.43 \pm 0.08) \times 10^{-3}$ | $(8.51 \pm 0.06) \times 10^{-3}$ |
| Alanine Tetra-Peptide ($d = 120$) | ✗ | ✗ | $1 \times 10^8$ | $-333.61 \pm 0.23$ | $(7.36 \pm 4.22) \times 10^{-2}$ | $(7.02 \pm 4.00) \times 10^{-2}$ | $(1.70 \pm 0.27) \times 10^{-2}$ | $(1.48 \pm 0.32) \times 10^{-2}$ |
| | ✓ | ✗ | $1 \times 10^8$ | $-333.99 \pm 0.00$ | $(2.16 \pm 0.05) \times 10^{-3}$ | $(1.93 \pm 0.04) \times 10^{-3}$ | $(1.24 \pm 0.02) \times 10^{-2}$ | $(9.55 \pm 0.14) \times 10^{-3}$ |
| | ✗ | ✓ | $1 \times 10^8$ | $-333.96 \pm 0.00$ | $(2.46 \pm 0.05) \times 10^{-3}$ | $(1.68 \pm 0.03) \times 10^{-3}$ | $(1.62 \pm 0.02) \times 10^{-2}$ | $(9.88 \pm 0.10) \times 10^{-3}$ |
| | ✓ | ✓ | $1 \times 10^8$ | $-334.00 \pm 0.00$ | $(2.17 \pm 0.09) \times 10^{-3}$ | $(1.57 \pm 0.09) \times 10^{-3}$ | $(1.43 \pm 0.03) \times 10^{-2}$ | $(9.65 \pm 0.27) \times 10^{-3}$ |
| Alanine Hexa-Peptide ($d = 180$) | ✗ | ✗ | $4 \times 10^8$ | $-531.48 \pm 0.21$ | $(2.56 \pm 0.37) \times 10^{-1}$ | $(2.52 \pm 0.43) \times 10^{-1}$ | $(3.75 \pm 0.04) \times 10^{-2}$ | $(3.68 \pm 0.09) \times 10^{-2}$ |
| | ✓ | ✗ | $4 \times 10^8$ | $-533.05 \pm 0.27$ | $(4.29 \pm 1.62) \times 10^{-2}$ | $(4.11 \pm 1.57) \times 10^{-2}$ | $(2.78 \pm 0.29) \times 10^{-2}$ | $(2.35 \pm 0.29) \times 10^{-2}$ |
| | ✗ | ✓ | $4 \times 10^8$ | $-533.06 \pm 0.02$ | $(1.28 \pm 0.10) \times 10^{-2}$ | $(1.39 \pm 0.12) \times 10^{-2}$ | $(3.19 \pm 0.63) \times 10^{-2}$ | $(2.20 \pm 0.02) \times 10^{-2}$ |
| | ✓ | ✓ | $4 \times 10^8$ | $-533.49 \pm 0.01$ | $(1.25 \pm 0.05) \times 10^{-2}$ | $(1.21 \pm 0.02) \times 10^{-2}$ | $(2.48 \pm 0.02) \times 10^{-2}$ | $(1.97 \pm 0.01) \times 10^{-2}$ |

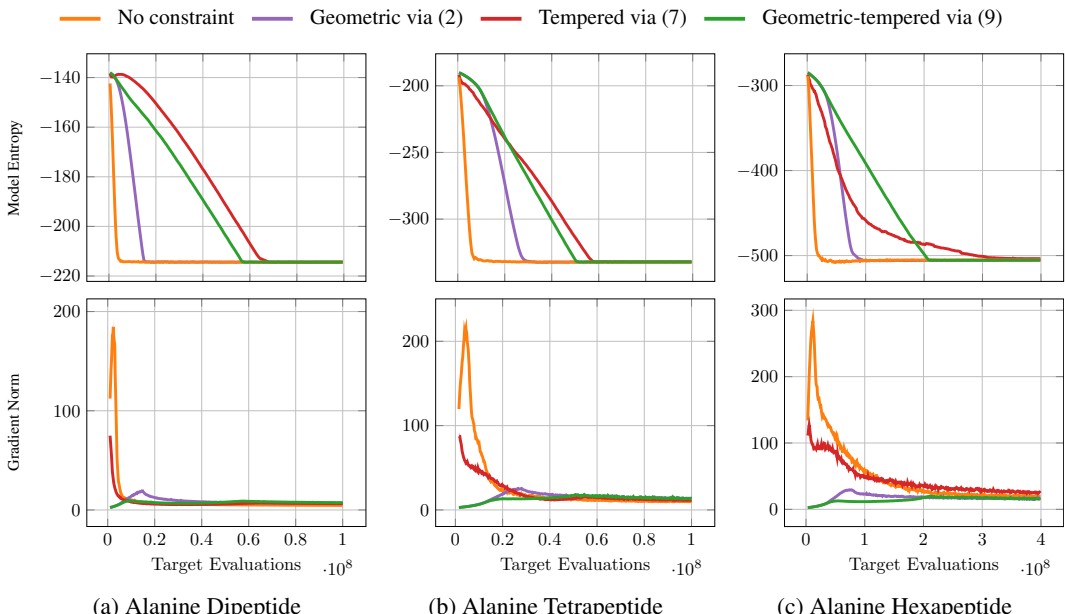

Figure 5: Effect of trust-region and entropy constraint on the model entropy (top row) and the gradient norm (bottom row) across different molecular systems.

**Ablation study for constraints.** Table 3 presents the performance of our method under different configurations, with the trust-region and entropy constraints selectively enabled or disabled. In addition to the alanine hexapeptide results shown in the main paper, we also report results for alanine dipeptide and alanine tetrapeptide. The absence of both constraints effectively corresponds to importance-weighted forward KL training. Considering the EUBO, which serves as a forward metric, it becomes clear that both constraints are necessary to achieve optimal performance. Variants of the method without the entropy constraint showed at least partial mode collapse, making the reverse ESS mostly incomparable. For this reason, in addition to reporting the EUBO, we also provide Ramachandran plot-based metrics to better assess mode collapse.

Figure 5 depicts the evolution of model entropy and the gradient norm (prior to clipping) during training across different systems. Training with only the entropy constraint yields an approximately linear decay of entropy for both alanine dipeptide and alanine hexapeptide. In the case of alanine hexapeptide, however, the entropy constraint is noticeably violated, likely due to the system's higher dimensionality and the pronounced discrepancy between the initial model distribution $q_0$ and the first intermediate distribution $q_1$. Larger system sizes also tend to increase the gradient norm, most prominently in alanine hexapeptide. The combination of the trust-region and entropy constraints produces the most stable gradient norms, while the approximately linear entropy decay indicates that the entropy constraint is effectively enforced, thereby enabling its practical application even in the case of alanine hexapeptide. By contrast, the trust-region constraint alone leads to a more rapid

entropy collapse, which reduces exploration and ultimately limits the algorithm's final performance in practice.

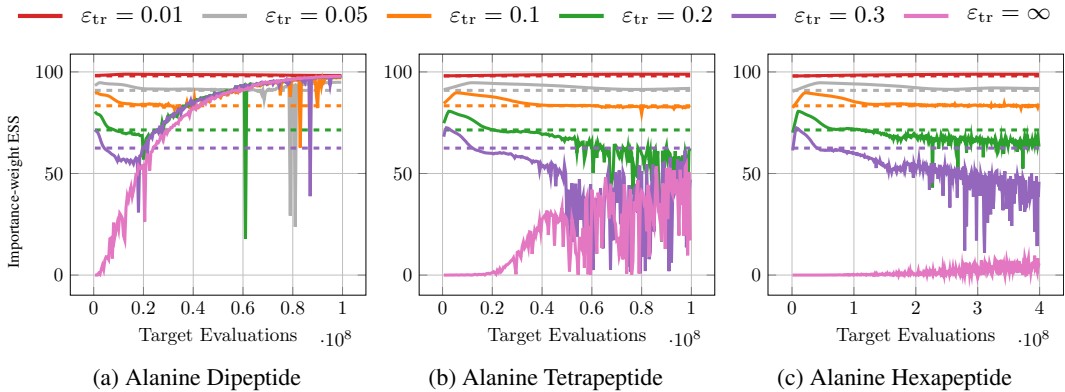

(a) Alanine Dipeptide  (b) Alanine Tetrapeptide  (c) Alanine Hexapeptide

Figure 6: Importance-weight variance between successive intermediate distributions, shown in terms of effective sample size (ESS), for different trust-region bounds and system sizes. Each trust-region bound $\varepsilon_{\mathrm{tr}}$ defines an approximate lower bound on the ESS, indicated by dashed lines.

**Ablation study on the trust-region bound.** Figure 6 illustrates the importance-weight variance of CMT across different trust-region bounds and system sizes, highlighting the approximate direct relationship between the trust-region bound and the variance of importance weights between consecutive intermediate distributions. Importance-weight variance is expressed in terms of effective sample size (ESS). In the absence of a trust-region constraint ($\varepsilon_{\mathrm{tr}} = \infty$), the ESS decreases with increasing system size. By contrast, finite trust-region bounds yield a substantially larger and more stable ESS, with the approximate lower bound on the ESS becoming increasingly well realized as the trust-region bound $\varepsilon_{\mathrm{tr}}$ decreases. Notably, this approximate lower bound is independent of the problem's dimensionality, a property that is empirically supported.

**Comparison to non-adaptive schedules**  To assess the importance of adaptivity, we performed two additional experiments on alanine hexapeptide comparing CMT to non-adaptive alternatives. We omit numerical results in this section, as they would not provide meaningful insight: both non-adaptive baselines exhibit substantially lower performance across all relevant metrics.

*Constant trust-region multiplier.* In the first experiment, we replace the adaptive trust-region schedule by a constant multiplier $\lambda_i = 43.3$ for all $i \in \mathbb{N}$ and remove the entropy constraint. The corresponding schedule is visualized in Figure 7. In practice, the non-adaptive variant performs significantly worse: training becomes unstable and exhibits substantial mode collapse, failing to adequately cover the target distribution.

*TA-BG with more annealing steps.* In the second experiment, we compare CMT to TA-BG with a fixed (non-adaptive) annealing schedule and varying numbers of annealing steps. TA-BG consists of a pre-training phase followed by a main training phase with annealing. To ensure a fair comparison with CMT (which uses $400$ annealing steps and a total of $800\,000$ gradient updates), we account for the fact that TA-BG uses $250\,000$ steps to pre-training and $550\,000$ steps to annealed training. Matching the total number of updates yields an effective number of annealing steps

$$\frac{550\,000}{250\,000 + 550\,000} \times 400 = 275.$$

We therefore tested both $275$ and $400$ annealing steps. Across both settings, TA-BG performs significantly worse.

**Energy histograms and TICA projections**  The target energy histograms for the different methods and systems are shown in Figure 8.

We further report the Wasserstein-2 distance relative to the ground-truth MD data, using $10^4$ samples for both the energy histograms and the TICA projections, as shown in Table 4.

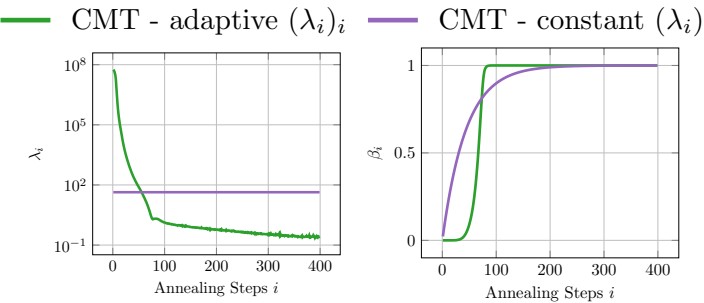

Figure 7: Visualization of a constant Lagrangian multiplier schedule $(\lambda_i)_i$ compared to an adaptive one using CMT on alanine hexapeptide. Both variants were performed without an entropy constraint.

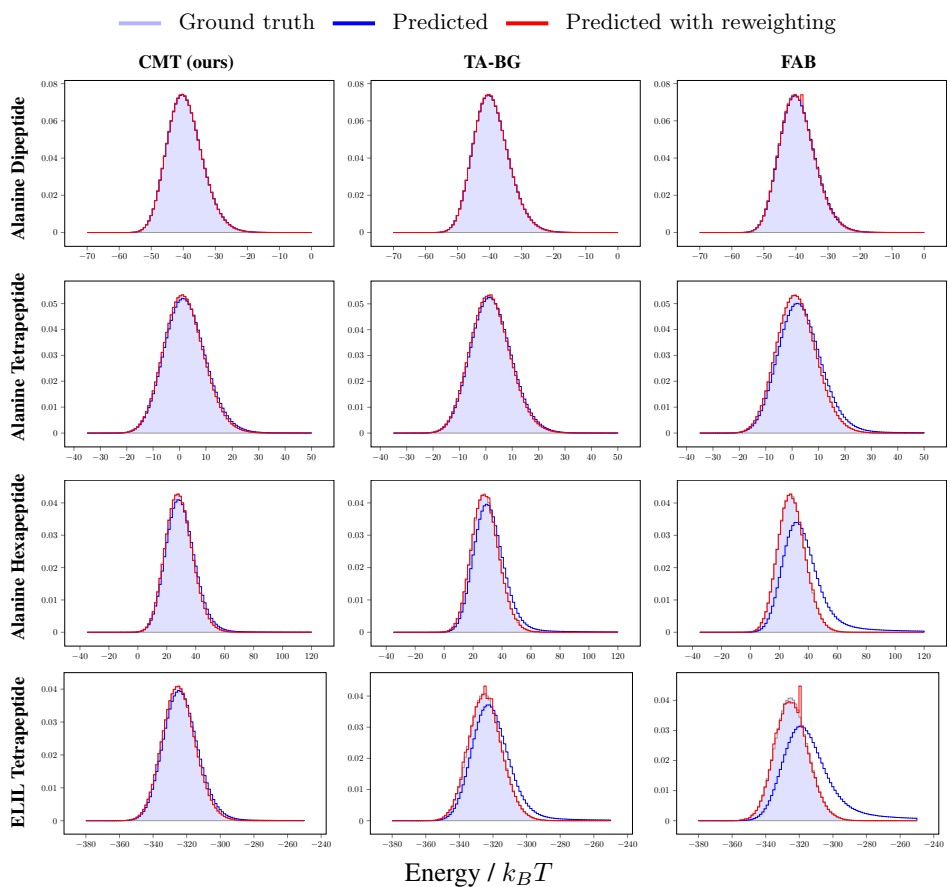

Figure 8: Comparison of target energy histograms shown as normalized density for different molecular systems (rows) and methods (columns).

We also ask the reader to focus on Figure 9, which shows several TICA projections of ELIL tetrapeptide. We chose this peptide because it most clearly illustrates the issue with using only $10^4$ samples for the Wasserstein metrics. Figure 9 qualitatively demonstrates that the Wasserstein-2 metric on the TICA projections using $10^4$ samples cannot reliably assess their quality. To further support this, we also report the TICA KL divergence between the MD TICA histogram and the predicted TICA histogram computed using $10^7$ samples on ELIL tetrapeptide in direct comparison to the TICA Wasserstein distance using $10^4$ samples (see Table 5).

Table 4: Wasserstein-2 distances for the Ramachandran plots (Ram-$\mathbb{T}$-$\mathcal{W}_2$, Ram-$\mathbb{T}$-$\mathcal{W}_2$ RW), energy histograms ($\mathcal{E}$-$\mathcal{W}_2$ RW), and TICA projections (TICA-$\mathcal{W}_2$, TICA-$\mathcal{W}_2$ RW), evaluated against the generated ground-truth MD data using $10^4$ samples. We report the Wasserstein-2 distance on the energy histograms only after reweighting (RW) to the target distribution, both to avoid domination by rare extreme target-energy values and to stay consistent with related work (Tan et al., 2025b;a). We also wish to emphasize that the Wasserstein metrics often exhibit large standard errors due to the comparatively small sample size used ($10^4$). For the same reason the Wasserstein-2 distance does not seem to be a reliable metric for comparing TICA projection quality. Further details are provided in Figure 9.

| SYSTEM | METHOD | RAM-$\mathbb{T}$-$\mathcal{W}_2$ ↓ | RAM-$\mathbb{T}$-$\mathcal{W}_2$ RW ↓ | $\mathcal{E}$-$\mathcal{W}_2$ RW ↓ | TICA-$\mathcal{W}_2$ | TICA-$\mathcal{W}_2$ RW |
|---|---|---|---|---|---|---|
| ALANINE DIPEPTIDE | FAB | $0.078 \pm 0.007$ | $0.085 \pm 0.012$ | $0.232 \pm 0.028$ | $0.025 \pm 0.004$ | $0.016 \pm 0.003$ |
| | TA-BG | $0.064 \pm 0.002$ | $0.073 \pm 0.002$ | $0.202 \pm 0.031$ | $\mathbf{0.015 \pm 0.001}$ | $0.021 \pm 0.003$ |
| | CMT (OURS) | $\mathbf{0.059 \pm 0.003}$ | $\mathbf{0.068 \pm 0.002}$ | $\mathbf{0.139 \pm 0.026}$ | $0.015 \pm 0.002$ | $\mathbf{0.012 \pm 0.001}$ |
| ALANINE TETRA- PEPTIDE | FAB | $0.617 \pm 0.010$ | $\mathbf{0.506 \pm 0.005}$ | $\mathbf{0.193 \pm 0.018}$ | $0.080 \pm 0.008$ | $\mathbf{0.029 \pm 0.005}$ |
| | TA-BG | $\mathbf{0.456 \pm 0.006}$ | $0.517 \pm 0.008$ | $0.263 \pm 0.053$ | $\mathbf{0.038 \pm 0.006}$ | $0.045 \pm 0.004$ |
| | CMT (OURS) | $0.492 \pm 0.006$ | $0.517 \pm 0.011$ | $0.235 \pm 0.035$ | $0.054 \pm 0.003$ | $0.048 \pm 0.007$ |
| ALANINE HEXA- PEPTIDE | FAB | $1.089 \pm 0.001$ | $1.055 \pm 0.015$ | $0.475 \pm 0.053$ | $0.183 \pm 0.005$ | $\mathbf{0.097 \pm 0.006}$ |
| | TA-BG | $0.838 \pm 0.006$ | $\mathbf{1.038 \pm 0.036}$ | $\mathbf{0.416 \pm 0.017}$ | $0.078 \pm 0.003$ | $0.117 \pm 0.026$ |
| | CMT (OURS) | $\mathbf{0.833 \pm 0.003}$ | $1.101 \pm 0.153$ | $0.817 \pm 0.314$ | $0.087 \pm 0.006$ | $0.114 \pm 0.012$ |
| ELIL TETRA- PEPTIDE | FAB | $0.813 \pm 0.011$ | $0.829 \pm 0.077$ | $1.133 \pm 0.418$ | $\mathbf{0.108 \pm 0.010}$ | $0.184 \pm 0.034$ |
| | TA-BG | $\mathbf{0.586 \pm 0.024}$ | $\mathbf{0.569 \pm 0.000}$ | $0.450 \pm 0.067$ | $0.150 \pm 0.013$ | $\mathbf{0.070 \pm 0.002}$ |
| | CMT (OURS) | $0.631 \pm 0.008$ | $0.628 \pm 0.048$ | $\mathbf{0.384 \pm 0.053}$ | $0.153 \pm 0.003$ | $0.084 \pm 0.011$ |

Table 5: TICA KL divergences (TICA KL and TICA KL RW) evaluated against the ground-truth MD data using $10^7$ samples, shown alongside the TICA Wasserstein distances computed with $10^4$ samples for comparison. Unlike the Wasserstein metric, which is less sensitive to fine details, the TICA KL with a larger sample size is more sensitive, as reflected in this table.

| SYSTEM | METHOD | TICA KL ↓ | TICA KL RW ↓ | TICA-$\mathcal{W}_2$ | TICA-$\mathcal{W}_2$ RW |
|---|---|---|---|---|---|
| ELIL TETRAPEPTIDE | FAB | $(6.82 \pm 0.41) \times 10^{-2}$ | $(4.37 \pm 0.37) \times 10^{-2}$ | $\mathbf{0.108 \pm 0.010}$ | $0.184 \pm 0.034$ |
| | CMT | $(8.58 \pm 0.22) \times 10^{-3}$ | $(1.05 \pm 0.05) \times 10^{-2}$ | $0.153 \pm 0.003$ | $\mathbf{0.084 \pm 0.011}$ |

**Ablation on different entropy bounds** We visualize the final performance of CMT on alanine hexapeptide for different entropy bounds $\varepsilon_{\mathrm{ent}}$ in Figure 10. While the optimal entropy bound $\varepsilon_{\mathrm{ent}}$ varies slightly between systems, we note that they are all in the same order of magnitude. Most hyperparameters of CMT remain constant across systems, making the entropy bound one of the few hyperparameters that requires some tuning. However, tuning $\varepsilon_{\mathrm{ent}}$ is straightforward in practice and described in detail in Appendix C.4.

**Comparison of CMT to MD** To compare CMT to MD, we analyze the mean absolute error (MAE) of the free energy profile along the $\phi$ dihedral angle, which is the slowest mode of the system, over the course of an MD simulation. We visualize the MAE as a function of target energy evaluations in Figure 11 and compare with the MAE obtained by CMT (using $1 \times 10^8$ target evaluations). MD requires approximately one order of magnitude more target evaluations to yield a similarly well-equilibrated free energy profile as CMT.

## C FURTHER DETAILS ON CONSTRAINED MASS TRANSPORT

In this section, we provide some additional details on constrained mass transport, including practical tips for hyperparameter selection, implementation and the connection to importance-weight variance.

### C.1 DUAL OPTIMIZATION IN PRACTICE

The concavity of the dual functions permits the use of any suitable nonlinear optimization algorithm. For one-dimensional dual optimization, we employ the bounded Brent method (Brent, 2013),

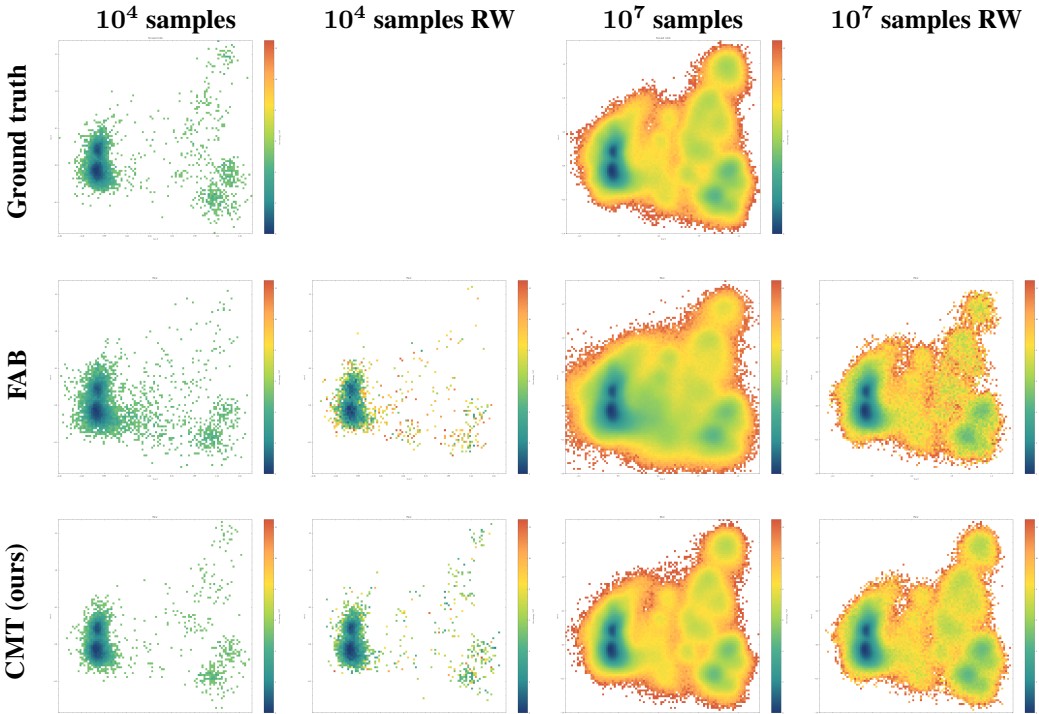

Figure 9: TICA projections for ground truth (MD), FAB, and CMT on ELIL tetrapeptide using $10^4$ and $10^7$ samples with and without reweighting to the target distribution. The TICA projection of CMT with $10^7$ samples is visibly more accurate, in contrast to FAB's, which appears slightly blurred or smoothed. For computational reasons, the Wasserstein metrics can only be computed using $10^4$ samples. The corresponding TICA projections are shown in the first column and clearly lack sufficient detail to accurately assess TICA projection quality.

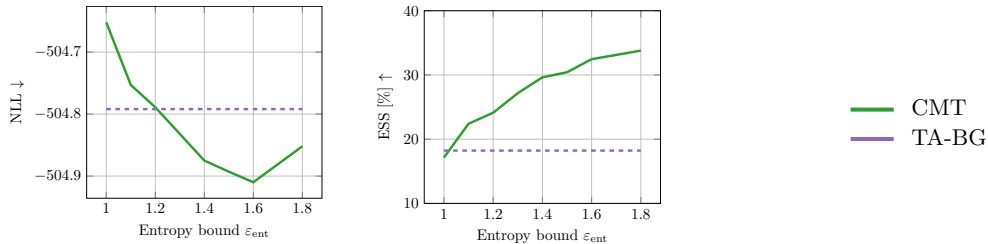

Figure 10: NLL and ESS performance with different entropy bounds for CMT on alanine hexapeptide.

implemented via `scipy.optimize.minimize_scalar` (Virtanen et al., 2020), which is the library's default 1D algorithm due to its robustness and efficiency. A minimal working example on how a Lagrangian multiplier is estimated is given in Code Example 1. For 2D duals, we use `scipy.optimize.minimize` with the L-BFGS-B algorithm (Zhu et al., 1997), one of SciPy's default quasi-Newton algorithms. There, we additionally passed the dual gradient function, which we obtained through automatic differentiation. Due to the constraints $\lambda, \eta \geq 0$, and to avoid numerical overflow, we bound both optimizers to stay within the interval $[0, 10^{10}]$. The method `scipy.optimize.minimize` requires an initial guess, which we set to $1 \times 10^{-20}$, a value chosen to be close to the lower bound.

We note that the pure dual optimization is very fast in practice. For alanine dipeptide, dual optimization time makes up approximately $0.01\,\%$ of the total training time. Details can be found in Appendix D.4.

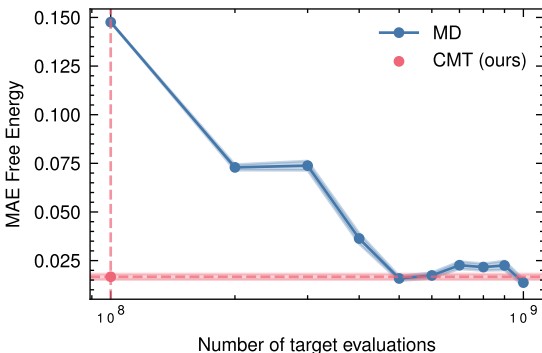

Figure 11: Comparison of CMT with molecular dynamics for alanine dipeptide. We show the mean absolute error of the free energy profile along the $\phi$ dihedral angle, which is the slowest mode of the system, as a function of the number of target evaluations.

## C.2 CMT WITH NORMALIZING FLOWS

Algorithm 1 allows for multiple variants. Algorithm 2 presents a specific instance of Algorithm 1, in which the KL-divergence is employed to update the normalizing flow model to the next intermediate distribution at each annealing step by adjusting its parameters. The dual function $g_{\varepsilon_{\mathrm{tr}},\varepsilon_{\mathrm{ent}}}^{(i+1)}$ is defined as in Equation (11), with Equation (16) providing the sample-based approximation. As is observable in Algorithm 2, the optimization of the Lagrangian multipliers and the computation of importance weights only add negligible computational and memory overhead, as they reuse the same buffer samples subsequently employed to update the flow model itself.

## C.3 CONNECTION TO IMPORTANCE-WEIGHT VARIANCE

In this section, we show that using the trust-region constraint yields an approximate lower bound for the effective sample size between any two consecutive distributions $q_i$ and $q_{i+1}$. This approximate lower bound only depends on $\varepsilon_{\mathrm{tr}}$.

This idea parallels common practice in sequential Monte Carlo (SMC). As stated by Chopin et al. (2023), line search on the $\chi^2$-divergence between consecutive distributions is commonly used to control the ESS. By restricting the $\chi^2$-divergence, SMC ensures that the importance weights do not degenerate, effectively enforcing a minimum ESS between intermediate distributions. Similarly, our trust-region constraint guarantees a conservative lower bound on ESS, providing a principled way to adaptively select the sequence of distributions.

The variance of the importance weights

$$\frac{q_{i+1}(x)}{q_i(x)} = \begin{cases} \frac{1}{\mathcal{Z}_{i+1}(\lambda_i)} \left( \frac{\tilde{p}(x)}{q_i(x)} \right)^{\frac{1}{1+\lambda_i}} & \text{with trust-region constraint (2)} \\ \frac{1}{\mathcal{Z}_{i+1}(\lambda_i,\eta_i)} \left( \frac{\tilde{p}(x)}{q_i(x)^{1+\eta_i}} \right)^{\frac{1}{1+\lambda_i+\eta_i}} & \text{with trust-region + entropy constraint (9)} \end{cases}$$

between two normalized consecutive distributions is closely connected to the effective sample size via

$$\mathrm{ESS}(q_i, q_{i+1}) = \frac{1}{1 + \mathrm{Var}_{q_i}\left( \frac{q_{i+1}(x)}{q_i(x)} \right)},$$

also explained in Appendix D.3. The relation $\mathrm{Var}_{q_i}(q_{i+1}(x)/q_i(x)) = \chi^2(q_{i+1}|q_i)$ (Chopin et al., 2020) and the well-known Taylor approximation $\chi^2(q_{i+1}|q_i) \approx 2D_{\mathrm{KL}}(q_{i+1}\|q_i)$ (Cover, 1999) lets us rewrite the effective sample size in terms of the Kullback-Leibler (KL) divergence between $q_{i+1}$ and $q_i$, yielding

$$\mathrm{ESS}(q_i, q_{i+1}) \approx \frac{1}{1 + 2D_{\mathrm{KL}}(q_{i+1}\|q_i)}$$

Code Example 1: Minimal working example of the dual optimization for objective (2).

```python
import numpy as np
import torch
from scipy.optimize import minimize_scalar

def estimate_log_Z(
    model_log_prob: torch.Tensor,
    target_log_prob: torch.Tensor,
    tr_mul: float,
) -> torch.Tensor:
    """Estimate log-partition function of next intermediate density"""
    log_N = torch.tensor(target_log_prob.shape[0]).log()
    log_iw = (target_log_prob - model_log_prob) / (1 + tr_mul)
    log_Z = torch.logsumexp(log_iw, dim=0) - log_N
    return log_Z

def find_best_kl_multiplier(
    model_log_prob: torch.Tensor,
    target_log_prob: torch.Tensor,
    eps_tr: float,
    max_multiplier: float = 1e10,
) -> float:
    """Finds the best Lagrangian multiplier by maximizing the dual"""
    # define dual function (dependent on Lagrangian multiplier)
    def dual(tr_mul: float):
        log_Z = estimate_log_Z(
            model_log_prob=model_log_prob,
            target_log_prob=target_log_prob,
            tr_mul=tr_mul,
        )
        dual_value = -(1 + tr_mul) * log_Z - tr_mul * eps_tr
        return dual_value.item()

    neg_dual = lambda mul: -dual(mul) # concave -> convex

    res = minimize_scalar(
        neg_dual,
        bounds=(0.0, max_multiplier),
        method="Bounded"
    )
    best_tr_mul = float(res.x)
    return best_tr_mul
```

as an approximation for the effective sample size. This approximation is justified under the assumption that $q_{i+1}$ is close to $q_i$, a condition that is satisfied by the design of the problem for a small trust-region bound $\varepsilon_{\mathrm{tr}} > 0$. Due to $q_{i+1}$ being the optimal solution to an objective with the constraint $D_{\mathrm{KL}}(q\|q_i) \leq \varepsilon_{\mathrm{tr}}$, the constraint must also hold for $q = q_{i+1}$ resulting in the approximate lower bound

$$\mathrm{ESS}(q_i, q_{i+1}) \gtrsim\approx \frac{1}{1 + 2\varepsilon_{\mathrm{tr}}} \tag{21}$$

for the effective sample size of the importance weights with equality in all but the last step.

This approximate lower bound justifies the use of Monte Carlo approximations in Section 3, helping to stabilize training independent of the problem's dimensionality. Empirical results on this property are provided in Appendix B.

---

**Algorithm 2** CMT with normalizing flows

---

**Require:** Initial normalizing flow $q_{\theta_0}$, unnormalized target density $\tilde{p}$, buffer size $N$, batch size $B$, chosen number of annealing steps $\tilde{I}$, number of iterations $K$ per annealing step, trust-region bound $\varepsilon_{\text{tr}}$, entropy bound $\varepsilon_{\text{ent}}$

  **for** $i \leftarrow 0, \ldots, \tilde{I} - 1$ **do**

    Draw $N$ samples $x_n \sim q_{\theta_i}(x)$, evaluate $q_{\theta_i}(x_n), \tilde{p}(x_n)$

    Initialize buffer $\mathcal{B}^{(i)} = (x_n, q_{\theta_i}(x_n), \tilde{p}(x_n))_{n=1}^{N}$

    Compute multipliers $\lambda_i, \eta_i = \arg\max_{\lambda, \eta \in \mathbb{R}^+} g_{\varepsilon_{\text{tr}}, \varepsilon_{\text{ent}}}^{(i+1)}(\lambda, \eta)$ using $\mathcal{B}^{(i)}$

    Define $q_{\theta_{i,0}} \coloneqq q_{\theta_i}$

    // Update flow to fit $q_{i+1}$:

    **for** $k \leftarrow 0, \ldots, K - 1$ **do**

      // Compute importance weights from $\mathcal{B}^{(i)}$:

      Retrieve mini-batch $b_k = (x_n, q_{\theta_i}(x_n), \tilde{p}(x_n))_{n=1}^{B}$ from $\mathcal{B}^{(i)}$

      Compute $(q_{i+1}(x_n))_{n=1}^{B}$ from $b_k$ and multipliers $\lambda_i, \eta_i$

      Compute importance weights $(w_n)_{n=1}^{B}$ as $w_n = q_{i+1}(x_n)/q_{\theta_i}(x_n)$ for $n = 1 \ldots B$

      // Update flow model:

      Define loss $l_k = -\frac{1}{B} \sum_{n=1}^{B} w_n \log q_{\theta_{i,k}}(x_n)$ // loss derived from $D_{\text{KL}}(q_{i+1} \| q_{\theta_{i,k}})$

      Update parameters from $\theta_{i,k}$ to $\theta_{i,k+1}$ using $l_k$

    Define $q_{\theta_{i+1}} \coloneqq q_{\theta_{i,K}}$

  **return** $q_{\theta_I} \approx p$

---

## C.4 GUIDANCE FOR HYPERPARAMETER SELECTION

Owing to the relationship between the trust-region constraint and the variance of importance weights, it may appear sensible to choose a very small trust-region bound. While doing so indeed improves the quality of the importance weights by lowering their variance, it simultaneously increases the number of intermediate annealing steps required for convergence. In practice, we observed this to constitute a trade-off between training stability and computational efficiency. For this reason, we adopted a fixed trust-region bound of $\varepsilon_{\text{tr}} = 0.3$, which consistently yielded the best performance across all systems under a fixed number of target evaluations.

Assuming the target entropy were known and the number of intermediate densities fixed to $\tilde{I} \in \mathbb{N}_+$, one could select the entropy bound as

$$\varepsilon_{\text{ent}} \geq \frac{H(q_0) - H(p)}{\tilde{I}},$$

where the inequality allows for the possibility that the entropy constraint becomes inactive earlier during training. In practice, however, the target entropy is rarely available. The design of our entropy constraint mitigates this issue by restricting the search domain for $\varepsilon_{\text{ent}}$ to $\mathbb{R}^+$. This restriction makes it straightforward to identify a suitable range for $\varepsilon_{\text{ent}}$. Due to the approximately linear entropy decay, empirically validated in Figure 5, the relationship between the iteration at which the entropy constraint becomes inactive and the entropy bound $\varepsilon_{\text{ent}}$ is likewise approximately linear, which makes tuning $\varepsilon_{\text{ent}}$ straightforward.

Across all molecular systems, suitable values for $\varepsilon_{\text{ent}}$ fall within the range $[0.8, 1.8]$. We aimed for the entropy constraint to become inactive after roughly half of the total training iterations, as this consistently yielded the best final performance. The entropy bound was tuned only to the first decimal place, with most other hyperparameters remaining fixed across systems.

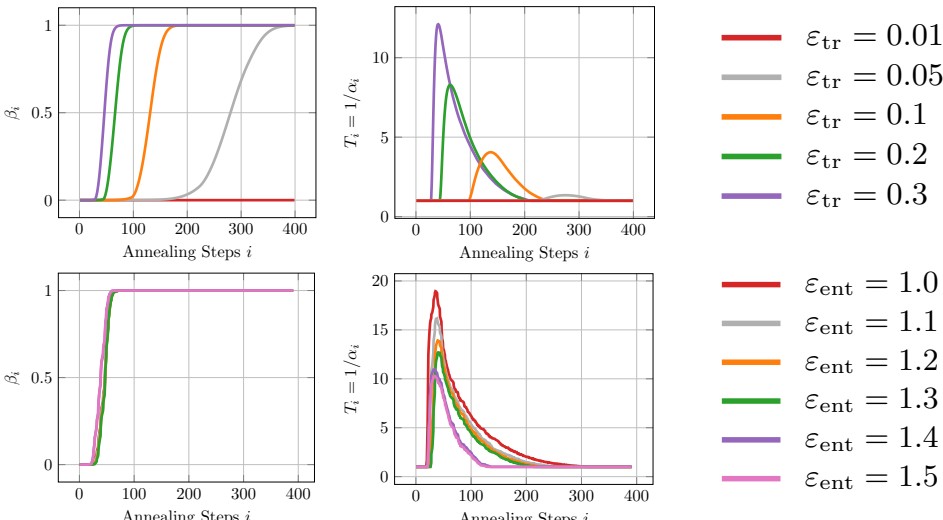

Figure 12: Visualization of $\beta_i$ and $T_i = 1/\alpha_i$ over the annealing steps $i$ on alanine hexapeptide for different trust-region bounds (first row) and entropy bounds (second row), each evaluated while the other bound is kept fixed ($\varepsilon_{\mathrm{tr}} = 0.3$ and $\varepsilon_{\mathrm{ent}} = 1.4$). Varying the trust-region bound visibly changes the convergence speed of both $(\beta_i)_i$ and $(T_i)_i$ along the annealing path, whereas varying the entropy bound seems to lead to similar behavior for $(\beta_i)_i$ but strongly affects the temperature behavior $(T_i)_i$. We further emphasize that the displayed $\beta_i$ does not accurately represent the actual "closeness" of $q_i$ to the target distribution $p$ during training. Each intermediate density is only approximated via $\hat{q}_i = \arg\min_q D(q_i, q)$, so the sequence $(\hat{q}_i)_i$ does not follow the analytical annealing path exactly. As a consequence, the calculated evolution of $\beta_i$ does not provide a reliable absolute measure of how close $q_i$ is to $p$ in terms of KL divergence.

## C.5 ANNEALING PATH VISUALIZATIONS

To complement Figure 1, we provide visualizations of the sequences $(\alpha_i)_i$ and $(\beta_i)_i$ along the annealing path

$$q_i \propto q_0^{\beta_i} \left( p^{\alpha_i} \right)^{1-\beta_i}$$

in Figure 12, illustrating the effects of different trust-region and entropy bounds.

## D EXPERIMENTAL SETUP

### D.1 ARCHITECTURE

Our normalizing flow architecture closely follows the one used in previous works (Midgley et al., 2022; Schopmans & Friederich, 2025; 2024). We represent the conformations of the studied molecular systems using internal coordinates based on bond lengths, angles, and dihedral angles.

We use 8 pairs of neural spline coupling layers based on monotonic rational-quadratic splines (Durkan et al., 2019). The splines map from $[0, 1]$ to $[0, 1]$ using 8 bins. We use a random mask to select transformed and conditioned dimensions in the first coupling of each pair, and the corresponding inverted mask for the second coupling. The dihedral angle dimensions are modeled with circular splines (Rezende et al., 2020) to respect their topology, with a random (fixed) periodic shift applied after each coupling layer. The parameter networks that calculate the spline parameters in each coupling are fully connected neural networks with hidden dimensions $[256, 256, 256, 256, 256]$ and ReLU activation functions. To capture their periodicity, dihedral angles $\psi_i$ are encoded as $(\cos\psi_i, \sin\psi_i)$ when passing them to the parameter network.

As the base distribution of the normalizing flow, we use a uniform distribution in $[0, 1]$ for the dihedral angles and a Gaussian truncated to $[0, 1]$ with mean $\mu = 0.5$ and standard deviation $\sigma = 0.1$ for the bond lengths and angles.

Table 6: Overview of the molecular systems and corresponding force field parametrization.

| NAME | SEQUENCE | NO. ATOMS | FORCE FIELD | CONSTRAINTS |
|---|---|---|---|---|
| ALANINE DIPEPTIDE | ACE-ALA-NME | 22 | AMBER FF96 WITH OBC1 IMPLICIT SOLVATION | NONE |
| ALANINE TETRAPEPTIDE | ACE-3·ALA-NME | 42 | AMBER99SB-ILDN WITH AMBER99 OBC IMPLICIT SOLVATION | HYDROGEN BOND LENGTHS |
| ALANINE HEXAPEPTIDE | ACE-5·ALA-NME | 62 | AMBER99SB-ILDN WITH AMBER99 OBC IMPLICIT SOLVATION | HYDROGEN BOND LENGTHS |
| ELIL TETRAPEPTIDE | GLU-LEU-ILE-LEU | 75 | AMBER99SB-ILDN WITH AMBER99 OBC IMPLICIT SOLVATION | HYDROGEN BOND LENGTHS |

We follow Schopmans & Friederich (2025) to map the internal coordinates to the range $[0, 1]$ of the spline transformations: Dihedral angles are divided by $2\pi$. Bond lengths and angles are shifted and scaled as $\eta'_i = (\eta_i - \eta_{i;\min})/\sigma + 0.5$, where $\eta_{i;\min}$ is obtained from a minimum energy structure after energy minimization. $\sigma$ was set to $0.07\,\mathrm{nm}$ for bond lengths and $0.5730$ for angle dimensions.

The studied molecular systems have two chiral forms (mirror images), L- and R-chirality, while in nature, one almost only finds the L-chirality. To constrain the generated molecular configurations to the L-chirality, we constrain the spline output ranges of the relevant dihedral angles (Schopmans & Friederich, 2025). Similarly, some atoms and groups (such as the hydrogen atoms in $CH_3$ groups) are permutation invariant in the force field energy parametrization, but have a preference in the ground truth molecular dynamics data due to very large barriers. Similarly to the chirality constraints, we constrain the splines such that only the permutation found in the ground truth data can be generated (Schopmans & Friederich, 2025).

### D.2    TARGET DENSITIES

The goal of all our experiments is to sample molecular systems at $300\,\mathrm{K}$. An overview of the studied molecular systems, including their force field parametrization, is given in Table 6. We explicitly note that the largest studied system, ELIL tetrapeptide, does not contain capping groups, in contrast to the other three systems.

The energy evaluations during training were performed with the OpenMM 8.0.0 (Eastman et al., 2024) CPU platform, using 18 workers in parallel.

Following previous work (Midgley et al., 2022; Schopmans & Friederich, 2025), we use a regularized energy function to avoid large van der Waals energies due to atom clashes:

$$E_{\mathrm{reg.}}(E) = \begin{cases} E, & \text{if } E \leq E_{\mathrm{high}}, \\ \log(E - E_{\mathrm{high}} + 1) + E_{\mathrm{high}}, & \text{if } E_{\mathrm{high}} < E \leq E_{\mathrm{max}}, \\ \log(E_{\mathrm{max}} - E_{\mathrm{high}} + 1) + E_{\mathrm{high}}, & \text{if } E > E_{\mathrm{max}}. \end{cases} \tag{22}$$

We set $E_{\mathrm{high}} = 1 \times 10^8$ and $E_{\mathrm{max}} = 1 \times 10^{20}$ (Midgley et al., 2022).

### DATASETS

We employed extensive molecular dynamics simulations to generate training and validation datasets with $10^6$ samples and test datasets with $10^7$ samples to calculate the metrics reported in Table 1. For each system, one molecular dynamics simulation was used to construct the training and validation dataset, and a separate simulation was performed for the test dataset. We used slightly updated simulation parameters compared to Midgley et al. (2022) and Schopmans & Friederich (2025) to obtain better-equilibrated data, which became especially relevant for alanine dipeptide, where we

noticed a small bias in the energy histogram of the MD data provided by Midgley et al. (2022) (Stimper et al., 2022a) and used by Schopmans & Friederich (2025).

1. For alanine dipeptide, molecular dynamics simulations were carried out using the OpenMM LangevinMiddle integrator with a time step of $1\,\mathrm{fs}$. The system was initially equilibrated for $200\,\mathrm{ns}$, after which a production run of $5\,\mu\mathrm{s}$ was performed.

2. For alanine tetrapeptide, alanine hexapeptide, and ELIL tetrapeptide, we conducted replica-exchange molecular dynamics (Sugita & Okamoto, 1999) simulations at temperatures of $300.0\,\mathrm{K}$, $332.27\,\mathrm{K}$, $368.01\,\mathrm{K}$, $407.60\,\mathrm{K}$, $451.44\,\mathrm{K}$, and $500.0\,\mathrm{K}$ using the OpenMM LangevinMiddle integrator with a time step of $1\,\mathrm{fs}$. Each replica was first equilibrated independently for $200\,\mathrm{ns}$ without exchanges, followed by an additional $200\,\mathrm{ns}$ of equilibration with exchanges. Production simulations of $2\,\mu\mathrm{s}$ per replica were then carried out. For dataset construction, we used the trajectory from the $300.0\,\mathrm{K}$ replica.

## D.3 METRICS

In this section, we present metrics used for both theoretical analysis and experimental evaluation. Unless stated otherwise, all metrics were evaluated using $10^7$ samples.

### D.3.1 TASK-AGNOSTIC METRICS

We begin with the forward metrics *evidence upper bound* (EUBO) and the related *negative log likelihood* (NLL). We then turn to the reverse metric *evidence lower bound* (ELBO), discuss its limitations in the context of Boltzmann generators, and conclude with the *effective sample size* (ESS) as a measure of sample quality. For details on these metrics, we refer to Blessing et al. (2024).

**Evidence upper bound (EUBO) and negative log-likelihood (NLL).** EUBO and NLL are both forward metrics, each equivalent to the forward KL divergence up to an additive constant. They are defined via

$$D_{\mathrm{KL}}(p\|q) = \underbrace{\mathbb{E}_{p(x)}\left[\log\frac{\tilde{p}(x)}{q(x)}\right]}_{\text{EUBO}} - \underbrace{\log\mathcal{Z}}_{\text{const. w.r.t. } q}$$

$$= \underbrace{-\mathbb{E}_{p(x)}\left[\log q(x)\right]}_{\text{NLL}} - \underbrace{H(p)}_{\text{const. w.r.t. } q}.$$

**Evidence lower bound (ELBO).** The ELBO is defined as

$$\mathrm{ELBO} = \mathbb{E}_{q(x)}\left[\log p(x) - \log q(x)\right],$$

where $q$ is the variational distribution and $p$ the target distribution. The ELBO provides a lower bound on $\log\mathcal{Z}$, while the EUBO provides an upper bound on $\log\mathcal{Z}$. Typically, both bounds are reported, as the difference between them, the EUBO–ELBO gap, gives a useful measure of how tight the variational approximation is. A small gap indicates that the variational distribution closely approximates the true target, while a large gap signals that the approximation may be poor.

**Important note on ELBO.** In our experiments, we found that the ELBO is not a reliable metric for comparing molecular Boltzmann generators. The Boltzmann distribution of molecular systems contains extremely steep regions due to the repulsive term of the Lennard-Jones interaction, which grows with $1/r^{12}$ when the distance $r$ between two atoms becomes small. Boltzmann generators typically cannot fully capture this behavior, resulting in a small number of samples with extremely small log importance weights. These outliers dominate the mean, making it unrepresentative of actual model performance. This issue does not affect metrics such as reverse ESS (and thus importance sampling performance), which are computed directly on the importance weights rather than their logarithms, nor forward metrics evaluated under the support of the ground truth data (e.g., EUBO). To mitigate this problem, we clamped the lowest $0.01\%$ of log importance weights to the highest observed value among them when computing the ELBO. While this approach works reasonably well for alanine dipeptide, it is insufficient for the other systems, which likely require even more aggressive clamping. However, we decided to keep the same threshold for all systems for consistency. Consequently, we caution against using the ELBO for quantitative comparisons on molecular Boltzmann distributions.

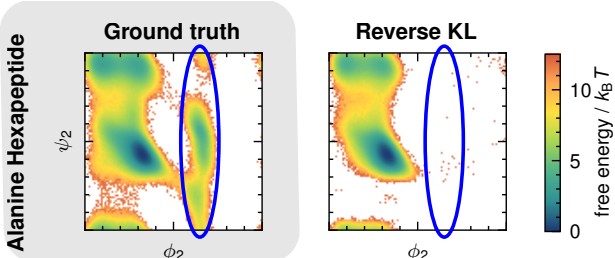

Figure 13: Illustration of mode collapse in Ramachandran plots. The figure highlights high-energy regions where mode collapse often occurs (e.g., under reverse KL training) but which remain important for accurately capturing of the peptide backbone distribution. Mode collapse is visible as the absence of density in these regions.

**Effective sample size (ESS).**    The ESS is defined as

$$\text{ESS}(a,b) = \frac{1}{1 + \text{Var}_{a(x)}\left[\frac{b(x)}{a(x)}\right]}, \quad a, b \in \mathcal{P}(\mathbb{R}^d).$$

Closely following the notation of Blessing et al. (2024), the reverse ESS

$$\text{ESS}(q,p) = \frac{\mathcal{Z}_r}{\mathbb{E}_{q(x)}\left[\left(\frac{\tilde{p}(x)}{q(x)}\right)^2\right]}, \quad \text{with} \quad \mathcal{Z}_r = \mathbb{E}_{q(x)}\left[\frac{\tilde{p}(x)}{q(x)}\right]$$

can be directly estimated via Monte Carlo using samples from the model $q$ and the unnormalized target $\tilde{p}$.

Following Midgley et al. (2022); Schopmans & Friederich (2025), for reverse ESS, we clipped the top $0.01\%$ importance-weights, setting them to the smallest value among them for numerical reasons. Furthermore, ESS is computed using the regularized energy function, defined in Equation (22).

Although forward ESS could be computed using samples from the target distribution, Schopmans & Friederich (2025) found it to be extremely sensitive to the chosen clipping threshold and prone to instability. Consequently, only the reverse ESS was used, even though it may not fully capture phenomena such as mode collapse.

### D.3.2    TASK-SPECIFIC METRICS

**Ramachandran plots.**    A Ramachandran plot visualizes the 2D log-density of the joint distribution of a pair of dihedral angles in a peptide's backbone. For more details, we refer to Schopmans & Friederich (2025). These plots are used to visualize a peptide's main degrees of freedom and are likely to show mode collapse if it occurs. A Ramachandran plot is effectively a histogram of the occurrence of dihedral angle values and is computed solely from model or ground-truth samples.

How mode collapse can be detected and where the high-energy density region is located in a Ramachandran plot is illustrated in Figure 13.

For alanine tetrapeptide, alanine hexapeptide, and the ELIL tetrapeptide, which contain multiple backbone dihedral angle pairs, we always show the pair exhibiting the most pronounced deviation from the ground truth, which is the same across methods. Among the four runs made per method in Figure 4, we selected the one with the lowest Ram KL value. For Figure 3, we always selected the run with the highest Ram KL value to illustrate that variations with fewer constraints are more likely to exhibit mode collapse.

**Ramachandran KL (Ram KL) and Ramachandran KL with reweighting (Ram KL w. RW).** Ram KL and Ram KL w. RW can be used to quantify deviation from the ground truth Ramachandran histograms. Following Midgley et al. (2022); Schopmans & Friederich (2025), we computed the forward KL divergence between the Ramachandran histograms from ground truth and model samples

Table 7: Computational cost (in hours) of FAB, TA-BG, and CMT on different molecular systems, reporting pre-training, training and total time. For all systems, we performed four parallel runs on a single node equipped with $4\times$ NVIDIA A100-SXM4-40GB and $2\times$ AMD EPYC Rome 7402 CPU with 48 cores in total.

| System | Method | Pre-training | Training | Total |
|---|---|---|---|---|
| Alanine Dipeptide | FAB | - | 18.1 h | 18.1 h |
| | TA-BG | 6.2 h | 16.9 h | 23.2 h |
| | CMT (ours) | - | 20.4 h | 20.4 h |
| Alanine Tetrapeptide | FAB | - | 19.5 h | 19.5 h |
| | TA-BG | 7.2 h | 24.3 h | 31.5 h |
| | CMT (ours) | - | 21.7 h | 21.7 h |
| Alanine Hexapeptide | FAB | - | 41.0 h | 41.0 h |
| | TA-BG | 22.1 | 43.7 h | 65.8 h |
| | CMT (ours) | - | 64.3 h | 64.3 h |
| ELIL Tetrapeptide | FAB | - | 56.6 h | 56.6 h |
| | TA-BG | 22.3 | 116.1 h | 138.4 h |
| | CMT (ours) | - | 138.2 h | 138.2 h |

(Ram KL). For this, we used $100 \times 100$ bins and $1 \times 10^7$ samples. Additionally, we also computed a reweighted version of the metric (Ram KL w. RW) where the model samples were first reweighted to the target distribution before generation of Ramachandran histograms. For the larger systems, where more than one Ramachandran plot exists, we reported the average.

**Ramachandran total variation (Ram TV)**   The Ram TV and Ram TV w. RW (with reweighting) distance complement the existing Ram KL and Ram KL w. RW metrics. While the KL-based metrics are asymmetric and computed with respect to the support of the ground-truth distribution, the total variation metrics are symmetric, allowing them to capture discrepancies both within and outside the support of the ground truth. This provides a more balanced assessment of how accurately the model represents the overall distribution. The total variation distance on Ramachandran plots, which are 2D discrete normalized histograms, is defined as

$$\text{TVD(P, Q)} = \frac{1}{2} \sum_{i,j} |P_{ij} - Q_{ij}| a_{ij},$$

where $P$ and $Q$ are the Ramachandran plots of the target $p$ and the model $q$, respectively. The factor $a_{ij}$ corresponds to the area of bin $(i, j)$; in our setting, all bins have equal area $a_{ij} = (2\pi/100)^2$.

**TICA KL**   Similarly to the Ramachandran KL, we also computed the TICA KL on the TICA histograms, using the range from the test dataset to define the domain.

**Wasserstein distance**   We report several Wasserstein-2 distances for evaluation: on the Ramachandran plots (Ram $\mathbb{T}$-$\mathcal{W}_2$), on the ground-truth energies of the predicted samples ($\mathcal{E}$-$\mathcal{W}_2$), and on the TICA vectors (TICA-$\mathcal{W}_2$). Details on the Wasserstein metric and its implementation can be found in Tan et al. (2025a). Loosely speaking, the Wasserstein-2 distance quantifies how much probability mass must be transported to transform one distribution into another. We apply it to the periodic dihedral angle pairs forming the Ramachandran plots by defining the distance on the resulting torus, thus respecting the periodicity of the angles. For details, see Tan et al. (2025a) (App. E.1). For computational reasons, we also only used $10^4$ samples, as the Wasserstein distance does not scale well to substantially larger sample sizes, which limits sensitivity to mode collapse in high-energy metastable regions.

## D.4   Computational cost

**Total training time**   To provide an indication of computational cost, we report extrapolated training times for FAB, TA-BG, and CMT across different molecular systems in Table 7. Because training

Table 8: Dual optimization time of CMT on different systems per annealing step and in total.

| SYSTEM | PER ANNEALING STEP | TOTAL |
|--------|-------------------|-------|
| ALANINE DIPEPTIDE | 53.31 ms | 10.6 s |
| ALANINE TETRAPEPTIDE | 52.48 ms | 10.50 s |
| ALANINE HEXAPEPTIDE | 61.25 ms | 24.50 s |
| ELIL TETRAPEPTIDE | 77.17 ms | 61.74 s |

Table 9: Number of flow parameters for each system. The number of parameters is completely determined by a molecular system's size, as the architecture is the same across all systems.

| | ALANINE DIPEPTIDE | ALANINE TETRAPEPTIDE | ALANINE HEXAPEPTIDE | ELIL |
|--|------------------|---------------------|---------------------|------|
| NUMBER OF PARAMETERS | 7 421 512 | 9 452 376 | 12 124 616 | 13 727 952 |

molecular Boltzmann generators is computationally expensive, the reported values are based on extrapolation and reflect pure training time only, excluding evaluation steps and other overhead.

Compared to Schopmans & Friederich (2025), we increased the total number of target evaluations and gradient descent steps for TA-BG to match those used for CMT, ensuring a fair comparison. While this modification improves the final performance of TA-BG, it also raises its computational cost to a level comparable to CMT. Since both CMT and TA-BG rely on similar training routines, their overall computational demands are generally comparable.

The comparatively high computational cost observed for CMT may be reduced through further optimization, for example by refining the loss function or decreasing the number of gradient descent steps and target evaluations. Its adaptive schedule and use of importance weights rather than resampling may also improve robustness when using smaller buffer sizes compared to TA-BG. We further emphasize, that our implementation represents a specific realization of a broader algorithmic framework, and additional efficiency improvements may be possible.

For FAB, we report training time using an updated number of target evaluations but retain the original number of gradient descent steps from (Midgley et al., 2022), thereby providing a low-computational-cost baseline.

The dual optimization time of all methods is negligible and summarized in Table 8.

**Inference time**  The inference time of all methods is identical, as they share the same underlying normalizing flow architecture, with its size varying only between systems but not between methods. Details on the architecture are provided in Appendix D.1.

### D.5  HYPERPARAMETERS

Hyperparameters play a crucial role in the performance of all models. Common hyperparameters include the choice of optimizer, learning rate, batch size, gradient steps, and weight decay. Below, we provide a description of the hyperparameters for each method, emphasizing any method-specific choices.

All experiments employed the Adam optimizer (Kingma & Ba, 2017). Our implementation builds on the Python packages *bgflow* (Noé & co workers, 2025), *nflows* (Durkan et al., 2020), and *PyTorch* (Paszke et al., 2019). The number of parameters in the normalizing flow architecture for each system is summarized in Table 9.

### CMT

We refer to Table 10 for the general and method-specific hyperparameters of CMT. The number of annealing steps $\tilde{I}$ is defined implicitly by the total number of gradient descent steps and the number of gradient descent steps per annealing step, and it is summarized in Table 11.

Table 10: Hyperparameter settings for CMT (general and method-specific) for all systems.

| | HYPERPARAMETERS | ALANINE DIPEPTIDE | ALANINE TETRAPEPTIDE | ALANINE HEXAPEPTIDE | ELIL |
|---|---|---|---|---|---|
| GENERAL | BATCH SIZE | 1000 | 1000 | 2000 | 2000 |
| | LEARNING RATE | $4 \times 10^{-5}$ | $5 \times 10^{-5}$ | $5 \times 10^{-5}$ | $5 \times 10^{-5}$ |
| | LR SCHEDULER | COSINE | COSINE | COSINE | COSINE |
| | GRADIENT DESCENT STEPS | 400 000 | 400 000 | 800 000 | 1 600 000 |
| | WEIGHT-DECAY | $1 \times 10^{-5}$ | $1 \times 10^{-5}$ | $1 \times 10^{-5}$ | $1 \times 10^{-5}$ |
| | LR LINEAR WARMUP STEPS | 1000 | 1000 | 1000 | 1000 |
| | MAX GRAD NORM | 100.0 | 100.0 | 100.0 | 100.0 |
| METHOD-SPECIFIC | TRUST-REGION BOUND | 0.3 | 0.3 | 0.3 | 0.3 |
| | ENTROPY BOUND | 0.8 | 1.8 | 1.4 | 0.7 |
| | BUFFER SIZE | 500 000 | 500 000 | 1 000 000 | 1 000 000 |
| | GRADIENT DESCENT STEPS PER ANNEALING STEP | 2000 | 2000 | 2000 | 2000 |

Table 11: Fixed number annealing steps $\tilde{I}$ for CMT for each system. The number is implicitly defined by the total number of gradient descent steps and the number gradient descent steps per annealing step. We keep it fixed to strictly control the computational budget for fair benchmarking.

| | ALANINE DIPEPTIDE | ALANINE TETRAPEPTIDE | ALANINE HEXAPEPTIDE | ELIL |
|---|---|---|---|---|
| NUMBER OF ANNEALING STEPS | 200 | 200 | 400 | 800 |

TA-BG

Table 12 summarizes the hyperparameters for the pre-training of TA-BG (Schopmans & Friederich, 2025) using the reverse KL divergence.

After pre-training, the temperature is annealed with a geometrically decaying temperature sequence and the hyperparameters summarized in Table 13. The TA-BG experiments on alanine dipeptide and alanine tetrapeptide used the geometric temperature annealing sequence

$$1200\,\mathrm{K} \to 1028.69\,\mathrm{K} \to 881.84\,\mathrm{K} \to 755.95\,\mathrm{K} \to 648.04\,\mathrm{K} \to 555.52\,\mathrm{K}$$
$$\to 476.22\,\mathrm{K} \to 408.24\,\mathrm{K} \to 349.96\,\mathrm{K} \to 300.00\,\mathrm{K} \to 300.00\,\mathrm{K}.$$

Including an additional finetuning step per temperature, TA-BG employs the temperature sequence

$$1200\,\mathrm{K} \to 1028.69\,\mathrm{K} \to 1028.69\,\mathrm{K} \to 881.84\,\mathrm{K} \to 881.84\,\mathrm{K} \to 755.95\,\mathrm{K}$$
$$\to 755.95\,\mathrm{K} \to 648.04\,\mathrm{K} \to 648.04\,\mathrm{K} \to 555.52\,\mathrm{K} \to 555.52\,\mathrm{K} \to 476.22\,\mathrm{K}$$
$$\to 476.22\,\mathrm{K} \to 408.24\,\mathrm{K} \to 408.24\,\mathrm{K} \to 349.96\,\mathrm{K} \to 349.96\,\mathrm{K} \to 300.00\,\mathrm{K} \to 300.00\,\mathrm{K}$$

on alanine hexapeptide. On the ELIL tetrapeptide, reverse KL pre-training suffers from mode-collapse at $1200\,\mathrm{K}$. Therefore, the temperature annealing starts at $3000\,\mathrm{K}$, resulting in the temperature sequence

$$3000.00\,\mathrm{K} \to 2573.09\,\mathrm{K} \to 2573.09\,\mathrm{K} \to 2573.09\,\mathrm{K} \to 2206.93\,\mathrm{K} \to 2206.93\,\mathrm{K}$$
$$\to 2206.93\,\mathrm{K} \to 1892.88\,\mathrm{K} \to 1892.88\,\mathrm{K} \to 1892.88\,\mathrm{K} \to 1623.52\,\mathrm{K} \to 1623.52\,\mathrm{K}$$
$$\to 1623.52\,\mathrm{K} \to 1392.49\,\mathrm{K} \to 1392.49\,\mathrm{K} \to 1392.49\,\mathrm{K} \to 1194.33\,\mathrm{K} \to 1194.33\,\mathrm{K}$$
$$\to 1194.33\,\mathrm{K} \to 1024.37\,\mathrm{K} \to 1024.37\,\mathrm{K} \to 1024.37\,\mathrm{K} \to 878.60\,\mathrm{K} \to 878.60\,\mathrm{K}$$
$$\to 878.60\,\mathrm{K} \to 753.57\,\mathrm{K} \to 753.57\,\mathrm{K} \to 753.57\,\mathrm{K} \to 646.34\,\mathrm{K} \to 646.34\,\mathrm{K}$$
$$\to 646.34\,\mathrm{K} \to 554.36\,\mathrm{K} \to 554.36\,\mathrm{K} \to 554.36\,\mathrm{K} \to 475.48\,\mathrm{K} \to 475.48\,\mathrm{K}$$
$$\to 475.48\,\mathrm{K} \to 407.81\,\mathrm{K} \to 407.81\,\mathrm{K} \to 407.81\,\mathrm{K} \to 349.78\,\mathrm{K} \to 349.78\,\mathrm{K}$$
$$\to 349.78\,\mathrm{K} \to 300.00\,\mathrm{K} \to 300.00\,\mathrm{K}.$$

FAB

The used hyperparameters for FAB (Midgley et al., 2022) can be found in Table 14. Furthermore, we used a step size of 0.05 for the Hamiltonian Monte Carlo (Duane et al., 1987) transitions. For details on the method and its hyperparameters, we refer to Midgley et al. (2022).

Table 12: Hyperparameter settings for TA-BG pre-training for all systems.

| | HYPERPARAMETERS | ALANINE DIPEPTIDE | ALANINE TETRAPEPTIDE | ALANINE HEXAPEPTIDE | ELIL |
|---|---|---|---|---|---|
| | TARGET TEMPERATURE | $1200\,\mathrm{K}$ | $1200\,\mathrm{K}$ | $1200\,\mathrm{K}$ | $3000\,\mathrm{K}$ |
| | BATCH SIZE | 256 | 256 | 512 | 512 |
| | LEARNING RATE | $1 \times 10^{-4}$ | $1 \times 10^{-4}$ | $1 \times 10^{-4}$ | $1 \times 10^{-4}$ |
| | LR SCHEDULER | COSINE | COSINE | COSINE | COSINE |
| GENERAL | GRADIENT DESCENT STEPS | 100 000 | 100 000 | 250 000 | 250 000 |
| | WEIGHT-DECAY | $1 \times 10^{-5}$ | $1 \times 10^{-5}$ | $1 \times 10^{-5}$ | $1 \times 10^{-5}$ |
| | LR LINEAR WARMUP STEPS | 1000 | 1000 | 1000 | 1000 |
| | MAX GRAD NORM | 100.0 | 100.0 | 100.0 | 100.0 |
| | NO. HIGHEST ENERGY VALUES REMOVED | 10 | 10 | 20 | 20 |

Table 13: Hyperparameter settings for TA-BG (general and method-specific) for all systems.

| | HYPERPARAMETERS | ALANINE DIPEPTIDE | ALANINE TETRAPEPTIDE | ALANINE HEXAPEPTIDE | ELIL |
|---|---|---|---|---|---|
| | BATCH SIZE | 2048 | 4096 | 2048 | 2048 |
| GENERAL | LEARNING RATE | $5 \times 10^{-6}$ | $1 \times 10^{-5}$ | $5 \times 10^{-6}$ | $5 \times 10^{-6}$ |
| | LR SCHEDULER | COSINE (PER ANNEALING STEP) | - | - | - |
| | GRADIENT DESCENT STEPS | 300 000 | 300 000 | 550 008 | 1 350 000 |
| METHOD-SPECIFIC | BUFFER SIZE | 7 440 000 | 7 440 000 | 15 111 111 | 22 400 000 |
| | BUFFER RESAMPLED TO | 2 000 000 | 2 000 000 | 2 000 000 | 22 400 000 |
| | GRADIENT DESCENT STEPS PER ANNEALING STEP | 30 000 | 30 000 | 30 556 | 45 000 |

Table 14: Hyperparameter settings of FAB (general and method-specific) for all systems.

| | HYPERPARAMETERS | ALANINE DIPEPTIDE | ALANINE TETRAPEPTIDE | ALANINE HEXAPEPTIDE | ELIL |
|---|---|---|---|---|---|
| | BATCH SIZE | 1024 | 1024 | 1024 | 2048 |
| | LEARNING RATE | $1 \times 10^{-4}$ | $1 \times 10^{-4}$ | $1 \times 10^{-4}$ | $2 \times 10^{-4}$ |
| | LR SCHEDULER | COSINE | COSINE | COSINE | COSINE |
| GENERAL | GRADIENT DESCENT STEPS | 50 000 | 50 000 | 50 000 | 25 000 |
| | WEIGHT-DECAY | $1 \times 10^{-5}$ | $1 \times 10^{-5}$ | $1 \times 10^{-5}$ | $1 \times 10^{-5}$ |
| | LR LINEAR WARMUP STEPS | 1000 | 1000 | 1000 | 1000 |
| | MAX GRAD NORM | 1000.0 | 1000.0 | 1000.0 | 1000.0 |
| METHOD-SPECIFIC | NO. INTERMED. DIST. | 8 | 8 | 8 | 16 |
| | NO. INNER HMC STEPS | 4 | 4 | 8 | 8 |

FORWARD AND REVERSE KL

This section reports the used hyperparameters for training with the forward KL divergence on MD data (Table 15) and the hyperparameters for training with the reverse KL divergence (Table 16). A description of how the MD data was obtained can be found in Appendix D.2.

Table 15: Hyperparameter settings of forward KL training using MD data for all systems.

| | HYPERPARAMETERS | ALANINE DIPEPTIDE | ALANINE TETRAPEPTIDE | ALANINE HEXAPEPTIDE | ELIL |
|---|---|---|---|---|---|
| GENERAL | BATCH SIZE | 1024 | 1024 | 1024 | 1024 |
| | LEARNING RATE | $5 \times 10^{-5}$ | $5 \times 10^{-5}$ | $5 \times 10^{-5}$ | $5 \times 10^{-5}$ |
| | LR SCHEDULER | COSINE | COSINE | COSINE | COSINE |
| | GRADIENT DESCENT STEPS | 100 000 | 100 000 | 120 000 | 140 000 |

Table 16: Hyperparameter settings of reverse KL training for all systems.

| | HYPERPARAMETERS | ALANINE DIPEPTIDE | ALANINE TETRAPEPTIDE | ALANINE HEXAPEPTIDE | ELIL |
|---|---|---|---|---|---|
| GENERAL | BATCH SIZE | 1024 | 1024 | 1024 | 1024 |
| | LEARNING RATE | $1 \times 10^{-4}$ | $1 \times 10^{-4}$ | $1 \times 10^{-4}$ | $1 \times 10^{-4}$ |
| | LR SCHEDULER | COSINE | COSINE | COSINE | COSINE |
| | GRADIENT DESCENT STEPS | 250 000 | 250 000 | 250 000 | 250 000 |
| | WEIGHT-DECAY | $1 \times 10^{-5}$ | $1 \times 10^{-5}$ | $1 \times 10^{-5}$ | $1 \times 10^{-5}$ |
| | LR LINEAR WARMUP STEPS | 1000 | 1000 | 1000 | 1000 |
| | MAX GRAD NORM | 100.0 | 100.0 | 100.0 | 100.0 |
| | NO. HIGHEST ENERGY VALUES REMOVED | 40 | 40 | 40 | 40 |

