# OpenReview forum: "Learning Boltzmann Generators via Constrained Mass Transport"
_ICLR.cc/2026/Conference — ICLR 2026 Poster_

### Official Review · Reviewer_dNiK · 2025-10-29

**Soundness:** 3
**Presentation:** 3
**Contribution:** 3
**Rating:** 6
**Confidence:** 4

**Summary:**

This paper introduces Constrained Mass Transport (CMT) for learning Boltzmann Generators via a sequence of constrained variational updates. The method proposes to minimize $\mathrm{KL}(q||p)$ subject to (1) a trust-region constraint $\mathrm{KL}(q||q_i)≤\varepsilon_{\text{tr}}$ and/or (2) a bound on entropy decay $H(q_i)−H(q)≤ \varepsilon_{\text{ent}}$. The authors derive closed-form optimal intermediate densities and show these induce geometric, tempered, or geometric-tempered annealing paths when (1), (2), or both are active, respectively. The authors demonstrate gains in performance under matched target evaluation budgets of the method on conformational sampling for Alanine peptide systems (dipeptide, tetrapeptide, and hexapeptide) and a new ELIL tetrapeptide system. CMT improves EUBO, ESS, and Ramachandran TV under matched target-evaluation budgets. Ablations indicate both constraints are needed to avoid mode collapse and stabilize ESS across steps.

**Strengths:**

1. The paper is well-written and easy to understand.

2. The method is conceptually clean. The paper leverages known existing ideas in reinforcement learning and cleverly adapt them to the sampling problem. The constrained objectives yield analytic intermediates and a unified view of annealing paths.

3. Strong empirical results across peptide systems and metrics under fixed energy budgets.

4. Useful ablations highlighting the complementary roles of the trust-region and entropy constraints.

**Weaknesses:**

1. It would strengthen the paper to report Energy-W2 and TICA-W2 alongside the current metrics. Rama-TV (and Torus-W2 / TICA-W2) are informative at the macro level, but a model can score well on such macro metrics while still missing fine-grained details e.g., bond-length, that Energy-W2 would better capture. TICA-W2 adds a complementary macro view tied to slow collective modes. Including these would provide a more complete picture of both global conformational coverage and local physical accuracy. [1]
2. Providing non-ML baselines (MD and SMC) for comparison under the same energy evaluation budget would better contextualize the contribution.
3. Authors provide an analysis for $\varepsilon_{\text{tr}}$ vs (ESS lower bounds) but not for $\varepsilon_{\text{ent}}​$. It would be interesting to see this.

[1] Tan, Charlie B., et al. "Scalable equilibrium sampling with sequential boltzmann generators." arXiv preprint arXiv:2502.18462 (2025).

**Questions:**

1. Have you tried Cartesian coordinates? Does CMT’s path construction help maintain overlap in higher dimensions and improve stability in training?
2. It would be interesting to see how this approach compares to sequential boltzmann generators that are trained on MD data to see if these energy-only methods are approached data-driven. [1]
3. Any special initialization for $q_0$​? How often do you observe divergence or mode collapse?
4. For alanine dipeptide and tripeptide, how does this compare to PITA (which is run on all-atom coordinates) [2]

[1] Tan, Charlie B., et al. "Scalable equilibrium sampling with sequential boltzmann generators." arXiv preprint arXiv:2502.18462 (2025).
[2] Akhound-Sadegh, Tara, et al. "Progressive Inference-Time Annealing of Diffusion Models for Sampling from Boltzmann Densities." arXiv preprint arXiv:2506.16471 (2025).

---

> ### Author Response · Authors · 2025-11-25
>
> We thank the reviewer for taking the time to review our work and for the many helpful comments and suggestions. We hope the following replies address the questions and concerns raised. Additionally, we uploaded a revised version of the manuscript that includes several changes as outlined in the sequel.
>
> ---
>
> > It would strengthen the paper to report Energy-W2 and TICA-W2 alongside the current metrics. Rama-TV (and Torus-W2 / TICA-W2) are informative at the macro level, but a model can score well on such macro metrics while still missing fine-grained details e.g., bond-length, that Energy-W2 would better capture. TICA-W2 adds a complementary macro view tied to slow collective modes. Including these would provide a more complete picture of both global conformational coverage and local physical accuracy.
> >
>
> We thank the reviewer for this suggestion. We have included these results in **Appendix E.4, Table 18** and **Figure 12** of the revised manuscript. Following the convention in recent literature and due to computational cost, we calculated all W2 metrics using $10^4$ samples (compared to $10^7$ samples used for other metrics). Using only $10^4$ samples leads to a large standard error on the TICA-W2 distances. Furthermore, as can be seen in **Figure 12**, $10^4$ samples are generally not enough to assess the TICA projection quality. To address this, we plan to include the total variation distance and KL divergence on the TICA projections using $10^7$ samples in the camera-ready version.
>
>
> ---
>
> > Providing non-ML baselines (MD and SMC) for comparison under the same energy evaluation budget would better contextualize the contribution.
> >
>
> We agree that a direct comparison with conventional sampling approaches is essential for demonstrating the superior sampling efficiency of our approach.
>
> To this end, we compared CMT with molecular dynamics (MD) simulations for alanine dipeptide. **Figure 14** of our revised manuscript reports the mean absolute error (MAE) of the free energy profile along the $\phi$ dihedral angle, which is the slowest mode of the system, as a function of the number of target energy evaluations. One can see that the MAE progressively improves over the length of the MD trajectory. However, MD requires approximately one order of magnitude more target evaluations to yield a similarly well-equilibrated free energy profile as CMT.
>
> This analysis highlights the superior sampling efficiency of CMT, not only compared to other variational ML baselines, but also classical MD.
>
>
> ---
>
> > Authors provide an analysis for $\epsilon_{\text{tr}}$ vs (ESS lower bounds) but not for $\epsilon_{\text{ent}}$. It would be interesting to see this.
> >
>
> We thank the reviewer for the suggestion. We attempted to derive a similar theoretical characterization for the entropy bound $\epsilon_{ent}$, but have not yet found a characterization analogous to the one for $\epsilon_{tr}$.  The connection between $\epsilon_{tr}$ and ESS is direct because the trust-region constraint bounds the KL divergence, which serves as a proxy for the variance of the importance weights (as detailed in Appendix B). We are currently investigating whether an alternative characterization can be established.
>
> ---
>
> > Have you tried Cartesian coordinates?
> >
>
> We have not used Cartesian coordinates in this work, as this represents a non-trivial change that would require fundamentally different architectures (e.g., SE(3)-equivariant flows) to handle rotational and translational symmetries. However, as noted in our Conclusion (Section 6), we view this as a very promising direction for future work, particularly for enabling the learning of transferable Boltzmann generators.
>
> ---
>
> > Does CMT’s path construction help maintain overlap in higher dimensions and improve stability in training?
> >
>
> Indeed, the trust-region constraint ensures distributional overlap regardless of the problem dimensionality. Empirically, Figure 6 confirms this: the importance-weight ESS remains high and stable across all three molecular systems $(d=60, 120, 180)$, respecting the approximate lower bound for small values of $\epsilon_{tr}$.
>
> Regarding stability, Figure 5 shows that the trust-region constraint leads to significantly smoother gradient norms compared to unconstrained variants, effectively mitigating the training instabilities observed in higher-dimensional settings.

---

> > ### Author Response · Authors · 2025-11-25
> >
> > > It would be interesting to see how this approach compares to sequential boltzmann generators that are trained on MD data to see if these energy-only methods are approached data-driven.
> > >
> >
> > Sequential Boltzmann Generators use a different architecture and operate in Cartesian coordinates, making a direct comparison difficult. However, we agree that benchmarking against data-driven methods is essential to contextualize performance. To that end, we included the 'Forward KL' baseline. This baseline trains the exact same architecture using MD data via Maximum Likelihood Estimation, therefore isolating the training objective (energy vs. data) from architectural differences, serving as the direct data-driven counterpart to our energy-based approach.
> >
> > ---
> >
> > > For alanine dipeptide and tripeptide, how does this compare to PITA (which is run on all-atom coordinates)
> > >
> >
> > PITA utilizes high-temperature training data and anneals the distribution of a proposal that was initially trained on this data. In contrast, our method is purely variational and does not rely on any simulation data. While related, our contribution thus ultimately solves a different and arguably harder task, making a direct comparison difficult.

---

### Official Review · Reviewer_QZLu · 2025-11-01

**Soundness:** 4
**Presentation:** 4
**Contribution:** 4
**Rating:** 8
**Confidence:** 4

**Summary:**

This paper describes a mathematically elegant approach for imposing both trust-region and entropy constraints on a mass transport problem.  This results in improved learning of the target distribution.

**Strengths:**

The paper is well written and motivated. The model is trained using energy evaluations, not data samples, which is a challenging and largely (for useful systems) unsolved problem.  The evaluations are sound and target larger than typical systems for this field (including a more complex peptide data set that will be contributed). Meaningful ablations are performed.

**Weaknesses:**

If there is a discussion of the computational efficiency of training and inference I can't find it - especially compared to other approaches.

**Questions:**

How does CMT balance exploration and convergence?

Will there be issues with exploration as systems scale up to proteins with rugged energy landscapes? Or will a uniform distribution on internal coordinates be sufficient?

---

> ### Author Response · Authors · 2025-11-25
>
> We thank the reviewer for taking the time to review our work and for the many helpful comments and suggestions. We hope the following replies address the questions and concerns raised. Additionally, we uploaded a revised version of the manuscript that includes several changes as outlined in the sequel.
>
> ---
>
> > If there is a discussion of the computational efficiency of training and inference I can't find it - especially compared to other approaches.
> >
>
> We kindly refer the reviewer to Appendix D.4 ‘Computational Cost’. There, we report the training times for our approach and TA-BG. The inference time is the same for all approaches since they use exactly the same normalizing flow architecture. We thank the reviewer and added a comment regarding the inference time to Appendix D.4 in the revised manuscript.
>
> ---
>
> > How does CMT balance exploration and convergence?
> >
>
> The trade-off between exploration and convergence can be controlled by varying the trust-region and entropy bounds $\varepsilon_{\text{tr}}, \varepsilon_{\text{ent}}$, respectively. Specifically, larger trust-region bounds accelerate the progression along the geometric annealing path, while larger entropy bounds accelerate the progression along the tempered annealing path. However, setting these values too large results in poor exploration / mode collapse.
>
> We further added a visualization (**Figure 9**) to the revised manuscript that shows the annealing schedules for different parameters $\varepsilon_{\text{tr}}$ and $\varepsilon_{\text{tr}}$, empirically confirming the behavior described above.
>
> ---
>
> > Will there be issues with exploration as systems scale up to proteins with rugged energy landscapes? Or will a uniform distribution on internal coordinates be sufficient?
> >
>
> We suspect that a uniform prior combined with an entropy constraint will be sufficient to learn the Boltzmann distribution of proteins, given enough computational resources. However, we consider such experiments to be outside the scope of the current work, as the tetrapeptide ELIL is already the largest and most complex system successfully studied with only target energy evaluations.

---

> > ### Comment · Reviewer_QZLu · 2025-11-25
> >
> > Having reviewed all comments from the reviewers and authors I remain enthusiastic about this paper. I suggest half a sentence be added to the main paper about the dual optimization only taking up 0.01% of training as myself and two other reviewers were expecting this to be more heavy weight.

---

> ### Author Response · Authors · 2025-11-26
>
> We thank the reviewer for the positive feedback. We added the wallclock times required for dual optimization in **Table 6** of the revised manuscript and included  a sentence in **Section 3**  of the main paper regarding the performance of the dual optimization.

---

### Official Review · Reviewer_jMV6 · 2025-11-02

**Soundness:** 4
**Presentation:** 4
**Contribution:** 4
**Rating:** 8
**Confidence:** 4

**Summary:**

The paper tackles sampling by learning normalizing flows along an annealing path defined by a sequence of constrained variational problems. At each step, it minimizes $\text{KL}(q\|p)$ to the target $p$ subject to two constraints: (i) a trust-region bound on $\text{KL}(q\|q_{\text{prev}})$ and (ii) a limit on entropy decay. The first constraint helps preserve overlap between iterates and the second helps avoid mass teleportation. They evaluate the approach on several high-dimensional molecular systems, including the ELIL tetrapeptide, which they introduce, and show consistent gains, particularly in effective sample size, over recent variational baselines.

**Strengths:**

The paper is clearly written and easy to follow. The motivation for combining trust-region and entropy constraints is well explained, and the authors do a good job connecting these ideas to the practical issues of maintaining overlap and avoiding mode collapse during annealing. While the individual components are known, their combination in this specific setting is original and well justified. The experimental section is strong and demonstrates clear empirical gains. Introducing the ELIL tetrapeptide as a new benchmark is also a nice contribution that could help future work in this area.

**Weaknesses:**

From a technical standpoint, the contribution is somewhat incremental. The main novelty is combining two existing constraints (trust-region and entropy regularization) within a single variational framework, rather than introducing a new theoretical ingredient. This does not detract from the paper given the strong empirical results and clear exposition.

**Questions:**

1) Can the authors clarify or provide intuition for why the entropy constraint mitigates mass teleportation? Is there a theoretical justification?

2) How robust is performance to the optimization of the Lagrange multipliers? Normalizing constant estimation can be tricky, which suggests possible issues, but one could also imagine that errors just "shift the annealing path" without hurting final performance.

3) How expensive is the constrained optimization relative to the rest of the training pipeline?

---

> ### Author Response · Authors · 2025-11-25
>
> We thank the reviewer for taking the time to review our work and for the many helpful comments and suggestions. We hope the following replies address the questions and concerns raised. Additionally, we uploaded a revised version of the manuscript that includes several changes as outlined in the sequel.
>
> ---
>
> > From a technical standpoint, the contribution is somewhat incremental. The main novelty is combining two existing constraints (trust-region and entropy regularization) within a single variational framework, rather than introducing a new theoretical ingredient. This does not detract from the paper given the strong empirical results and clear exposition.
> >
>
> We sincerely thank the reviewer for acknowledging our strong empirical results. Regarding the technical contribution, we would like to highlight that, to the best of our knowledge, our work is the first to connect trust-region and entropy regularization with annealing paths. We believe this insight can benefit other methods that currently rely on geometric annealing, for example, Sequential Monte Carlo methods, by enabling the use of schedulers derived from these constraints.
>
> ---
>
> > Can the authors clarify or provide intuition for why the entropy constraint mitigates mass teleportation? Is there a theoretical justification?
> >
>
> Mass teleportation typically involves a sudden shift to a disjoint high-probability region. By explicitly constraining the entropy decay rate $H(q_i) - H(q) \le \epsilon_{ent}$, we force the distribution to remain "spread out", ensuring it maintains overlap with the previous distribution rather than "snapping" to a distant mode. In response, we have updated Figure 1 to illustrate this intuition more clearly.
>
> While we have no rigorous proof that the entropy constraint prevents mass teleportation, Proposition 2.2 shows that the entropy constraint introduces a Lagrangian multiplier $\eta$ into the optimal density form: $q_{i+1} \propto \tilde{p}^{1/(1+\eta)}$. This $\eta$ acts as an adaptive temperature parameter; if the model attempts to collapse too quickly (violating the entropy constraint), $\eta$ increases, effectively flattening (tempering) the target density to enforce a smoother transition.
>
> ---
>
> > How robust is performance to the optimization of the Lagrange multipliers? Normalizing constant estimation can be tricky, which suggests possible issues, but one could also imagine that errors just "shift the annealing path" without hurting final performance.
> >
>
> Indeed, while estimation errors in $Z_{i+1}$ and $q_{i+1}$ cause deviations from the optimal annealing path between $q_0$ and $p$, each estimated intermediate density $\hat q_{i+1}$can be viewed as the starting point of a slightly shifted annealing path toward the target. Importantly, as the reviewer has already noted, this does not degrade the final performance, since the method does not suffer from error accumulation across successive iterations $i$.
>
> Finally, we note that optimizing the Lagrangian dual does not require estimating the normalization constant $\mathcal{Z}$ of the target density. Instead, it relies on the normalization constants of the intermediate targets
> $\mathcal{Z}\_{i+1}$, which are easier to estimate, as the trust-region constraint ensures the proposal distribution remains close to the intermediate target. This can also be seen empirically in Figure 6: the importance-weight ESS to the next intermediate target remains high and stable, allowing accurate estimation of $\mathcal{Z}_{i+1}$. We agree that this is an important detail to understand why our algorithm is scalable across dimensions, and we added a clarifying statement to our manuscript.
>
> ---
>
> > How expensive is the constrained optimization relative to the rest of the training pipeline?
> >
>
> The constrained optimization step involves solving a two-dimensional concave problem, whose computational cost is negligible compared to the rest of the training pipeline: E.g. for alanine dipeptide, the average dual optimization time per annealing step takes $53.31 \mathrm{ms}$. With 200 annealing steps, this takes roughly $10.6\mathrm{s}$, which corresponds to $\approx0.01$ \% of the total cost of the pipeline. We added this information to **Appendix C.1** and **D.4** of our revised manuscript.

---

### Official Review · Reviewer_hjaT · 2025-11-03

**Soundness:** 3
**Presentation:** 2
**Contribution:** 2
**Rating:** 6
**Confidence:** 2

**Summary:**

The work “Learning Boltzmann Generators via Constraned Mass Transport” deal with the learning of a Boltzmann distritbution throuhg the minimization of the reverse KL divergence, sometimes named Boltzmann Generators. The authors proposes a variational approach based on a annealing procedure, where the optimization scheme is further constrain by (i) contraining the entropy between successing temperatures and (ii) the divergence between two successives temperature. The author shows that their result present better results than other approach, and show in their experiment that their protocol doesn’t seem to suffer from mode collapse.

**Strengths:**

The papers proposes a reasonably simple protocol to optimize over the KL divergence adding further constraints. The authors derive the optimal distribution of their optimization constraint, very similar to the classical reference Neal 2001. Their experiments seem quite convincing that their method seem to perform better than the others without suffering from mode collapse.

**Weaknesses:**

To my opinion, the authors do not provide a clear explanation on why their method should work best w.r.t other annealing scheme. At least, it was not clearly explained why constraining the entropy and the KL divergence should tackle the mass teleportation. It would be a strong added value to have a analytical or intuitive explanation, possibly on a toy example, on why this new optimization schemes avoid this problem.

**Questions:**

1. I think it would be a strong added value to be able to apply the method on a toy model suffering from mass teleportation to understand how the optimization scheme managed to mitigate this problem.

    2. It is not discuss if the additional constraint add make the method slower w.r.t. other approaches, and if yes how much slower ? That would be a valuable information.

    3.It looks to me the manuscript overlooked some references which seems closely related to the present work, which might be worth discussing, concerning respectively mode collapse when using KL divergence,( Soletskyi, Gabrié, Loureiro, B. (2025). “A theoretical perspective on mode collapse in variational inference”.); the use of normalizing flows (Gabrié, Rotskoff, Vanden-Eijnden, E. PNAS (2022). “Adaptive Monte Carlo augmented with normalizing flows”); and the use of a well adapted annealing path with a different protocol to avoid mass teleportation (“Fast training and sampling of Restricted Boltzmann Machines”, Béreux et al, ICLR 2025).

---

> ### Author Response · Authors · 2025-11-25
>
> We thank the reviewer for taking the time to review our work and for the many helpful comments and suggestions. We hope the following replies address the questions and concerns raised. Additionally, we uploaded a revised version of the manuscript that includes several changes as outlined in the sequel.
>
> ---
>
> > To my opinion, the authors do not provide a clear explanation on why their method should work best w.r.t other annealing scheme. At least, it was not clearly explained why constraining the entropy and the KL divergence should tackle the mass teleportation.
> >
>
> Mass teleportation typically involves a sudden shift to a disjoint high-probability region. By explicitly constraining the entropy decay rate $H(q_i) - H(q) \le \epsilon_{ent}$, we force the distribution to remain "spread out", ensuring it maintains overlap with the previous distribution rather than "snapping" to a distant mode. Moreover, Proposition 2.2 shows that he entropy constraint introduces a Lagrangian multiplier $\eta$ into the optimal density form: $q_{i+1} \propto \tilde{p}^{1/(1+\eta)}$. This $\eta$ acts as an adaptive temperature parameter; if the model attempts to collapse too quickly (violating the entropy constraint), $\eta$ increases, effectively flattening (tempering) the target density to enforce a smoother transition that avoids this sudden shift to a disjoint high-probability region.
>
> We intended to provide this intuition in Figure 1, but we agree that the original visualization was cluttered and potentially confusing. We have therefore included an updated figure which contrasts the pathological jumps of the geometric annealing path with the smooth, mode-bridging transport enabled by our geometric-tempered AP.
>
> ---
>
> > I think it would be a strong added value to be able to apply the method on a toy model suffering from mass teleportation to understand how the optimization scheme managed to mitigate this problem.
> >
>
> We hope that our new Figure 1 addresses this concern, and we would be more than happy to provide additional clarifications/experiments if needed.
>
> ---
>
> > It is not discuss if the additional constraint add make the method slower w.r.t. other approaches, and if yes how much slower ? That would be a valuable information.
> >
>
> We appreciate the reviewer's query regarding the computational overhead introduced by our constraints. We address this based on two possible interpretations of the comparison:
>
> **Comparison to baselines:** If the reviewer wishes to compare the total runtime of our approach to the baselines (FAB, TA-BG), we refer to the detailed analysis provided in **Appendix D.4, Table 5**. This table shows that the total training time for CMT is comparable to the state-of-the-art baseline TA-BG.
>
> **Overhead of the constrained optimization.** If the reviewer is interested in the runtime cost added solely by the dual optimization necessary for the adaptive constraints (compared to a hypothetical fixed annealing schedule):
> The computational overhead is negligible. Since the dual optimization requires maximizing a concave function over one or two variables, and it reuses the expensive energy evaluations already performed for the model update, the cost is minimal.
>
> For example, on alanine dipeptide, the average dual optimization time per annealing step takes $77.995 \times 10^{-3} s$. With 400 annealing steps, this totals roughly $32s$, which corresponds to approximately $\mathbf{0.01\%}$ of the total computational cost of the pipeline. We added this information to **Appendix C.1** of our revised manuscript.
>
> ---
>
> > It looks to me the manuscript overlooked some references which seems closely related to the present work, which might be worth discussing, concerning respectively mode collapse when using KL divergence,( Soletskyi, Gabrié, Loureiro, B. (2025). “A theoretical perspective on mode collapse in variational inference”.); the use of normalizing flows (Gabrié, Rotskoff, Vanden-Eijnden, E. PNAS (2022). “Adaptive Monte Carlo augmented with normalizing flows”); and the use of a well adapted annealing path with a different protocol to avoid mass teleportation (“Fast training and sampling of Restricted Boltzmann Machines”, Béreux et al, ICLR 2025).
> >
>
> We thank the reviewer for pointing us to these references. They are indeed very related, and we apologize for missing them in the initial submission. We have included them in the updated manuscript.

---

### Official Review · Reviewer_VDn6 · 2025-11-03

**Soundness:** 2
**Presentation:** 3
**Contribution:** 2
**Rating:** 2
**Confidence:** 4

**Summary:**

This paper tackles the problem of sampling from unnormalized densities with Boltzmann generators. It proposes Constrained Mass Transport (CMT), which introduces trust-region and entropy constraints to effectively learn an adaptive annealing schedule that interpolates between the prior and the target distributions. The proposed method is evaluated on several molecular systems (Alanine dipeptide, tetrapeptide, hexapeptide, and ELIL tetrapeptide).

**Strengths:**

* The idea of learning an adaptive schedule for the annealing path is conceptually interesting and well-motivated.
* The paper is well-written and easy to follow. The mathematical details are clearly presented, and the proofs appear sound as far as I could verify.

**Weaknesses:**

* **Unclear specification of the number of annealing steps.** The number of annealing steps ($I$) appears to be a hyperparameter but is never explicitly stated. From Table 7, it seems to be roughly 200 steps for ALDP (400k total / 2000 per step ≈ 200).
    * This number is quite large, and it is unclear whether the reported gains over TA-BG stem from the adaptive schedule itself or simply from using many more steps. Ablations comparing CMT to TA-BG with an equivalent number of steps would clarify this.
    * Additionally, the conceptual role of $I$ is somewhat unclear to me, once the trust-region parameter $\epsilon_{\mathrm{tr}}$ is introduced. Since $\epsilon_{\mathrm{tr}}$ already constrains the KL divergence between successive distributions and, as a result,  implicitly determines how many steps are required, fixing both $I$ and $\epsilon_{\mathrm{tr}}$ doesn’t make sense to me.

* **Metric design and presentation are confusing and seem cherry-picked.** Several of the most representative metrics appear only in the appendix (Table 2). In the main table, the authors report only Rama TV. Why was this specific metric chosen? Even if one chooses to report total variation, shouldn’t the reweighted Rama TV be reported, given that the process is not explicitly pinned to the target distribution? Similarly, the reweighted Rama $\mathbb{T}\text{-}W_2$ metric is omitted. The rationale for which metrics appear in the main results versus the appendix should be clarified.

* **Mismatch in optimization budgets across methods.** The number of gradient steps differs substantially between CMT and TA-BG (see Tables 7 vs 8–9 in the appendix). While the authors mention equalizing the number of target evaluations, comparisons should also be made with the same number of gradient steps. Otherwise, it isn’t clear how this plays a role in the reported performance improvements.

* **Hyperparameter selection seems ad-hoc and system-specific.** The trust-region bound $\epsilon_{\mathrm{tr}}$ is fixed globally, but the entropy bound $\epsilon_{\mathrm{ent}}$ varies per molecular system with no principled tuning rule. Appendix C.4 even notes that entropy tuning is difficult. This per-system manual adjustment undermines the claim of a “learned schedule.” The authors should provide clearer guidance or automatic criteria for selecting $\epsilon_{\mathrm{ent}}$, and ablations showing its impact on training stability and performance.

* **Limited intuition about the learned schedule.** The paper emphasizes that CMT learns the annealing path, yet it never visualizes or analyzes what that path looks like. Providing intuition about the learned schedule and explaining why it may be advantageous over those in TA-BG would make the results much more interpretable.

* **No analysis of $\log Z$ along the annealing path.** Since the learned intermediates depend on Monte Carlo estimates of the partition function, it is important to assess how accurate those estimates are throughout the path. The paper only reports overall $\log Z$ bounds (ELBO/EUBO) after training, with no diagnostics of these estimates or variance of the weights along the trajectory. Showing how $\log Z$ (or ESS) evolves over the annealing path would substantiate the claim of the papers and make it easier to evaluate the effectiveness of the proposed adaptive schedule.

* **Missing justification for how CMT avoids mass teleportation.** My understanding of “mass teleportation” follows from [1], where the linear interpolation path results in an ill-conditioned and exploding vector field, which results from the fact that the relative weight of the modes isn’t preserved. However, by this definition, I do not understand how constraining the entropy decay helps. Can the authors clarify what they mean by mass teleportation?


[1] Máté, B., & Fleuret, F. (2023). Learning Interpolations between Boltzmann Densities. arXiv preprint arXiv:2301.07388. https://arxiv.org/abs/2301.07388

**Questions:**

See questions/weaknesses above.

---

> ### Author Response · Authors · 2025-11-25
>
> We thank the reviewer for their time and helpful feedback. We hope the responses below address all concerns. A revised manuscript with the corresponding changes has also been uploaded.
>
> ---
>
> > **Unclear specification of the number of annealing steps.**
> The number of annealing steps ($I$) appears to be a hyperparameter but is never explicitly stated. From Table 7, it seems to be roughly 200 steps for ALDP (400k total / 2000 per step ≈ 200). This number is quite large, and it is unclear whether the reported gains over TA-BG stem from the adaptive schedule itself or simply from using many more steps. Ablations comparing CMT to TA-BG with an equivalent number of steps would clarify this.
> >
>
> We thank the reviewer for noting the missing annealing-step values $I$; these are now included for all systems in **Section D.5**.
>
> We tested TA-BG on alanine hexapeptide with $275\ (=400 \times \text{training-iterations} / \text{pre-training-iterations})$ and 400 annealing steps, using the same number of gradient updates as CMT. Except for small improvements on ESS, TA-BG did not increase in performance (**Section E.3**).
>
> To better illustrate the advantage of our adaptive schedule, we added an ablation in **Section E.3,** comparing to a fixed geometric schedule with the same compute budget. The fixed schedule yields lower ESS and shows clear mode collapse in the Ramachandran plots.
>
>
>
> ---
>
> > **Conceptual role of the number of annealing steps** $I$.
> Additionally, the conceptual role of $I$ is somewhat unclear to me, once the trust-region parameter is introduced. Since already constrains the KL divergence between successive distributions and, as a result, implicitly determines how many steps are required, fixing both and doesn’t make sense to me.
> >
>
> While using Lagrangian multipliers as a stopping criterion is possible (as $\lambda=\eta=0$ implies satisfied constraints), we used a fixed $I$ to strictly control the computational budget for fair benchmarking (in our case, target evaluations). We further found that continuing the training process even after constraints become inactive serves as beneficial fine-tuning, improving the final model quality.
>
> We agree that introducing $I$ may be confusing, so we added clarifying details to **Section 5.1** and **Section D.5**.
>
> ---
>
> > **Metric design and presentation are confusing and seem cherry-picked.**
> Several of the most representative metrics appear only in the appendix (Table 2). In the main table, the authors report only Rama TV. Why was this specific metric chosen? Even if one chooses to report total variation, shouldn’t the reweighted Rama TV be reported, given that the process is not explicitly pinned to the target distribution? Similarly, the reweighted Rama $\mathbb{T}$-$\mathrm{W}_2$ metric is omitted. The rationale for which metrics appear in the main results versus the appendix should be clarified.
> >
>
> We report non-reweighted metrics in the main table because they more directly compare the learned proposal distributions. When proposals are close to the target, reweighted metrics become nearly identical across methods and thus less informative. We use Ram TV rather than Ram KLD because TV is symmetric and better reflects the bidirectionality required when matching generated and target Boltzmann distributions.
>
> We agree that this was not explained clearly in the initial submission and will clarify it in the camera-ready version.
>
> Furthermore, we added the reweighted Ramachandran $\mathbb{T}$-$\mathrm{W}_2$ metric to **Table 2**. Similar to the TICA plots in **Figure 12**, we want to emphasize that Wasserstein distances with $10^4$ samples are not a very reliable metric. For this reason, we refer to the other metrics, computed on $10^7$ samples. Using $10^7$ samples is computationally unfeasible for the Wasserstein distance.
>
> ---
>
> ---
>
> > **Mismatch in optimization budgets across methods.**
> The number of gradient steps differs substantially between CMT and TA-BG (see Tables 7 vs 8–9 in the appendix). While the authors mention equalizing the number of target evaluations, comparisons should also be made with the same number of gradient steps. Otherwise, it isn’t clear how this plays a role in the reported performance improvements.
> >
>
> We agree that equalizing the number of gradient steps, in addition to the target evaluations, is necessary to ensure a strictly fair comparison.
>
> Accordingly, we reran the TA-BG comparison with equalized gradient steps. The updated results, now in **Section E.2**, show that our method still outperforms TA-BG. We will update the camera-ready version to instead report TA-BG with an equal number of gradient steps for all systems.

---

> > ### Author Response · Authors · 2025-11-25
> >
> > > **Hyperparameter selection seems ad-hoc and system-specific.**
> > The trust-region bound is fixed globally, but the entropy bound varies per molecular system with no principled tuning rule. Appendix C.4 even notes that entropy tuning is difficult. This per-system manual adjustment undermines the claim of a “learned schedule.” The authors should provide clearer guidance or automatic criteria for selecting , and ablations showing its impact on training stability and performance.
> > >
> >
> > We thank the reviewer and agree that the “Guidance for hyperparameter selection” section lacked clarity regarding tuning the entropy bound. We have revised and extended this section and added an ablation on alanine hexapeptide (Appendix E.5) to illustrate the sensitivity of this hyperparameter. Although system-specific, we consistently found the best performance when the entropy constraint became inactive after roughly half of the training. Because the entropy bound and the iteration at which the constraint becomes inactive are related in an approximately linear manner, tuning $\epsilon_\mathrm{ent}$ was straightforward in practice. Moreover, while most other hyperparameters remained unchanged across molecular systems, adjusting the entropy bound alone was sufficient to obtain strong performance. The ablation in **Appendix E.5, Figure 13,** further shows that CMT outperforms TA-BG and FAB across a range of different entropy bounds.
> >
> > ---
> >
> > > **Limited intuition about the learned schedule.**
> > The paper emphasizes that CMT learns the annealing path, yet it never visualizes or analyzes what that path looks like. Providing intuition about the learned schedule and explaining why it may be advantageous over those in TA-BG would make the results much more interpretable.
> > >
> >
> > We agree with the reviewer that visualizing the learned schedule is valuable for making the results more interpretable. As such, we have added the corresponding visualizations to the revised manuscript in **Figure 9**, visualizing the annealing path using varying trust-region and entropy bounds, and **Figure 10**, comparing with a constant trust-region multiplier. We also updated Figure 1 to provide a clearer and more intuitive visualization of the different annealing paths.
> >
> > ---
> >
> > > **No analysis of**  $\log \mathcal{Z}$ **along the annealing path.**
> > Since the learned intermediates depend on Monte Carlo estimates of the partition function, it is important to assess how accurate those estimates are throughout the path. The paper only reports overall $\log \mathcal{Z}$ bounds (ELBO/EUBO) after training, with no diagnostics of these estimates or variance of the weights along the trajectory. Showing how  $\log \mathcal{Z}$ (or ESS) evolves over the annealing path would substantiate the claim of the papers and make it easier to evaluate the effectiveness of the proposed adaptive schedule.
> > >
> >
> > We kindly refer the reviewer to Figure 2 (b), where we provide a comparison for the ESS evolution along the annealing path between different constraints, denoted by $\text{ESS}(\hat q_{i-1},q_i)$. The intermediate normalization constants $\log \mathcal{Z_i}$ are much easier to estimate compared to the normalization constant of the target $\log \mathcal{Z}$, given that the trust-region bound ensures sufficient overlap between successive intermediates. This is shown in Figure 2 (b) and Figure 6, where the trust-region constraint leads to high ESS values.
> >
> > ---
> >
> > > **Missing justification for how CMT avoids mass teleportation.**
> > My understanding of “mass teleportation” follows from [1], where the linear interpolation path results in an ill-conditioned and exploding vector field, which results from the fact that the relative weight of the modes isn’t preserved. However, by this definition, I do not understand how constraining the entropy decay helps. Can the authors clarify what they mean by mass teleportation?
> > >
> >
> > By “mass teleportation”, we refer to the failure mode of geometric annealing in which probability mass abruptly shifts to regions where the current intermediate has **negligible density**, blocking effective transport. This differs from the definition of Máté & Fleuret (2023), who focus on preserving relative weights (often termed “mode switching” [2]). We agree that the current version lacks clarity and will add a more concrete definition in the camera-ready version.
> >
> > The entropy constraint mitigates this by modifying the path to $q_{i} \propto q_{0}^{1-\beta_{i}}(\tilde{p}^{\alpha_{i}})^{\beta_{i}}$, where $\alpha_i$ acts as an inverse temperature (Theorem 2.4). By constraining entropy decay, we force $\alpha_i$ to remain small in early steps. This flattens the target term $\tilde{p}^{\alpha_i}$, broadening its support to create a probability "bridge" that ensures overlap between disjoint modes. As shown in Figure 1, this prevents the sudden appearance of modes (teleportation) observed in standard geometric schedules.
> >
> > [2] Phillips et al, Particle Denoising Diffusion Sampler

---

### Author Response · Authors · 2025-11-28
**Global Response to All Reviewers**

---

We sincerely thank the reviewers for taking their time reviewing our paper and for the many helpful comments and suggestions. In response, we have added several new experiments and clarifications to the revised manuscript. For ease of access, these additions are included in *Appendix E: Rebuttal*.

---

A summary of the changes is provided below:

- **Experiments:**
    - Added a visualization of the learned schedules (Appendix E.1).
    - Compared against TA-BG with an increased number of gradient updates (matching those of CMT) on alanine tetrapeptide and alanine hexapeptide (Appendix E.2).
    - Compared our method to non-adaptive annealing schedules (Appendix E.3).
    - Provided visualizations of the energy histograms (with and without reweighting) for all systems, and additionally reported the energy and TICA Wasserstein distance (Appendix E.4).
    - Ablated different values of the entropy bound to demonstrate the robustness of our method (Appendix E.5).
    - Compared our method to classical MD on alanine dipeptide (Appendix E.6).
- **Clarifications:**
    - Revised Figure 1 to more clearly illustrate how the geometric-tempered annealing path avoids the pathological behavior of the standard geometric annealing path.
    - Added runtime measurements for the dual optimization, showing that it is negligible relative to the rest of the training pipeline.

---

Kind regards, the authors

---

> ### Author Response · Authors · 2025-12-03
>
> We would like to leave a final comment regarding the review by reviewer VDn6. We have addressed all concerns and questions raised by the reviewer and added a number of additional ablation experiments to our manuscript. We have further listed all relevant changes in the global comment for convenience. Thus, we believe that the original score of the review does not reflect the quality and impact of our work. We hope this can be taken into account for the final decision.

---

### Meta-Review · Area_Chair_88Do · 2026-01-05

**Summary:**

All the reviewers agree that the paper approaches an important problem of regularizing the probability path for sampling from unnormalized density. The shared concerns are the following:
- The main goal of the proposed method is to construct a probability path that avoids the mass teleportation, i.e. that the resulting sequence of marginals can be simulated via a vector field which satisfies the corresponding continuity equation. However, the authors do not prove that their method achieves this. Thus, the introduced constraints on the KL divergence and the entropy can be considered as useful heuristics.
- Several reviewers pointed out that the method combines several existing approaches, which limits its novelty.
- The reported metrics are different from the conventional evaluation in the field. Furthermore, as pointed out by reviewers QZLu and VDn6, the empirical evaluation oftentimes misses the computational budget used and compares methods with different budgets.

**Reviewer Concerns:**

The authors clarified the questions and partially addressed concerns about the empirical evaluation. Notably, the common concern about demonstrating that the method prevents mass teleportation remained unaddressed (the authors merely reiterated the intuition behind the proposed method).

Furthermore, the additional results support the validity of the critique by reviewer VDn6. In particular, the proposed method does not perform well on the metrics suggested by VDn6. Finally, all the updated results are added to the appendix, while nothing prevented the authors from updating the results in the main body of the paper. I consider this an unaddressed concern.

**Reviewer Scores:**

The main concerns about the empirical evaluation raised by VDn6 remain unaddressed. I envision two potential scenarios for the discussion: an active discussion between the reviewers might lead to a decrease in scores due to the issues raised by VDn6, whereas a passive discussion would likely lead reviewers to maintain their scores at the current high-variance evaluation.

---

### Decision · Program_Chairs · 2026-01-26

Accept (Poster)